# Systematic analysis of the molecular and biophysical properties of key DNA damage response factors

Joshua R Heyza[1], Mariia Mikhova[1,2], Aastha Bahl[1], David G Broadbent[1,3,4], Jens C Schmidt[1,5]*

[1]Institute for Quantitative Health Science and Engineering, Michigan State University, East Lansing, United States; [2]Department of Biochemistry and Molecular Biology, Michigan State University, East Lansing, United States; [3]College of Osteopathic Medicine, Michigan State University, East Lansing, United States; [4]Department of Physiology, Michigan State University, East Lansing, United States; [5]Department of Obstetrics, Gynecology, and Reproductive Biology, Michigan State University, East Lansing, United States

*For correspondence:
schmi706@msu.edu

Competing interest: The authors declare that no competing interests exist.

**Abstract** Repair of DNA double strand breaks (DSBs) is integral to preserving genomic integrity. Therefore, defining the mechanisms underlying DSB repair will enhance our understanding of how defects in these pathways contribute to human disease and could lead to the discovery of new approaches for therapeutic intervention. Here, we established a panel of HaloTagged DNA damage response factors in U2OS cells which enables concentration-dependent protein labeling by fluorescent HaloTag ligands. Genomic insertion of HaloTag at the endogenous loci of these repair factors preserves expression levels and proteins retain proper subcellular localization, foci-forming ability, and functionally support DSB repair. We systematically analyzed total cellular protein abundance, measured recruitment kinetics to laser-induced DNA damage sites, and defined the diffusion dynamics and chromatin binding characteristics by live-cell single-molecule imaging. Our work demonstrates that the Shieldin complex, a critical factor in end-joining, does not exist in a preassembled state and that relative accumulation of these factors at DSBs occurs with different kinetics. Additionally, live-cell single-molecule imaging revealed the constitutive interaction between MDC1 and chromatin mediated by its PST repeat domain. Altogether, our studies demonstrate the utility of single-molecule imaging to provide mechanistic insights into DNA repair, which will serve as a powerful resource for characterizing the biophysical properties of DNA repair factors in living cells.

## Editor's evaluation

This manuscript reports valuable tools and data to study DNA repair and its regulation in life cells by generating and validating cell lines with Halo-tag fusions to the chromosomal genes encoding ATM, NBS1, MDC1, RNF168, RNF169, 53BP1, RIF1, SHLD3, REV7, SHLD2, SHLD1, and DNA-PKcs. The data establish the utility of most of the tools but remain incomplete. Conclusions from the kinetic analysis would benefit from more validation by genetic experiments and the single particle tracking analysis offers more potential for analysis.

## Introduction

Genomic DNA is constantly exposed to a variety of endogenous and exogenous agents that induce DNA double strand breaks (DSBs). These agents include reactive oxygen species, metabolic

byproducts, and environmental carcinogens. If left unrepaired, DSBs can lead to the loss of genetic information, chromosome rearrangements, mutations, and telomere fusions that can result in the development of a wide variety of human diseases including cancer, immunodeficiencies, neurological syndromes, and premature aging disorders. In mammalian cells two main pathways exist for resolving DNA DSBs, non-homologous end joining (NHEJ) and homologous recombination repair (HR) (*Scully et al., 2019*). In addition, cells have evolved a complex signaling network that senses DNA lesions, called the DNA damage response (DDR). HR requires a sister chromatid to serve as a template for high-fidelity DSB repair and is limited to S- and G2-phases of the cell cycle. In contrast, NHEJ functions independently of a repair template by ligating together broken DNA ends often leading to genomic insertions and deletions. Because of the toxicity of persistent, unrepaired DNA DSBs, NHEJ functions ubiquitously throughout G1-, S-, and G2-phases of the cell cycle. While cells balance the use of NHEJ and HR particularly in S- and G2-phase, NHEJ-dependent repair represents the main pathway resolving the bulk of DSBs arising in human cells either through rapid recruitment of the core NHEJ complex (DNA-PKcs, KU70, KU80, XLF, XRCC4, and LIG4) at unresected DNA ends or after 53BP1-Shieldin mediated end fill-in of previously resected DSBs (*Setiaputra and Durocher, 2019*).

The DDR is critical for regulating whether DSBs are repaired via HR or NHEJ. DDR action encompasses three specific steps: DNA break detection, DDR signal amplification, and recruitment of processing enzymes and repair effectors. DSB detection can be carried out both by the Ku70/Ku80 heterodimer to directly recruit DNA-PKcs as well as ATM kinase and the MRN (Mre11, Rad50, and NBS1) complex which associate with DNA breaks leading to phosphorylation of H2AX at S139 (γH2AX) by ATM (*Ciccia and Elledge, 2010*). Next, DDR signal amplification is mediated by MDC1, RNF8, and RNF168 (*Doil et al., 2009*; *Kolas et al., 2007*; *Lukas et al., 2004*). MDC1 directly binds γH2AX through its C-terminal BRCT domains leading to recruitment of RNF8 and K63-linked polyubiquitylation of linker histone H1 (*Mailand et al., 2007*; *Thorslund et al., 2015*). RNF8-mediated histone ubiquitylation is critical for subsequent downstream recruitment of DSB repair effector proteins, such as 53BP1 and BRCA1, to the chromatin regions flanking DSBs. K63-linked ubiquitination serves as a docking site for RNF168 which further amplifies the DDR signal via its E3 ubiquitin ligase activity by depositing monoubiquitin marks onto H2A at K13 and K15 (H2AK13ubK15ub) (*Mattiroli et al., 2012*).

The critical step in repair pathway choice is the recruitment of either BRCA1 to promote HR or 53BP1 to facilitate NHEJ. One key regulator of this choice is RNF169 which binds H2AK13ubK15ub deposited by RNF168 (*Kitevski-LeBlanc et al., 2017*). RNF169 is a negative regulator of 53BP1 accumulation at DSBs and promotes BRCA1 recruitment (*An et al., 2018*; *R. Menon et al., 2019*). 53BP1 binds to H2AK15ub in addition to H4K20me2 which serves as the basis for recruitment of 53BP1 to DSBs (*Wilson et al., 2016*). 53BP1 binding in turn triggers recruitment and assembly of the 53BP1-RIF1-Shieldin complex where 53BP1 associates with RIF1 which in turn binds to SHLD3, REV7, SHLD2 and SHLD1 (*Dev et al., 2018*; *Gupta et al., 2018*; *Noordermeer et al., 2018*; *Figure 1A*). The Shieldin complex directly binds to ssDNA mediated by the OB-fold domains of SHLD2 (*Dev et al., 2018*; *Gao et al., 2018*; *Noordermeer et al., 2018*). Another important regulator of DSB repair pathway choice is the extent of end resection at the DNA break. Resection is carried out by several nucleases including CtIP and the MRN complex, Exo1, and DNA2 (*Zhao et al., 2020*). Extensive end resection is required for HR while limited resection favors NHEJ. One important function of the Shieldin complex is to protect DNA breaks by competing with HR factors for ssDNA binding and inhibiting access to exonucleases to prevent further resection. In addition, the Shieldin complex promotes end fill-in by recruiting Polα/Primase via the CST (CTC1, STN1, and TEN1) complex through an interaction between SHLD1 and CTC1 (*Mirman et al., 2022*). By preventing resection and promoting end fill-in the Shieldin complex mediates the formation of DNA ends compatible with NHEJ.

While we have extensive knowledge of the proteins involved in DNA repair, their genetic interactions, and biochemical activities, how their dynamic recruitment to DNA breaks controls repair pathway choice and preserves genome integrity is poorly understood. Importantly, absolute protein abundance, the mechanism by which repair factors search for breaks (e.g. 3D-diffusion, chromatin sampling/scanning), the dynamics of DNA break binding, and the sequence of repair factor recruitment to DSBs are critical determinants of repair pathway choice. In this study we have generated a collection of cell lines that express 12 HaloTagged DNA repair factors from their endogenous genomic loci. The tagged proteins encompass a variety of functions including DSB detection (ATM and NBS1), DDR signal amplification (MDC1 and RNF168) and repair effector recruitment (RNF169, 53BP1, RIF1,

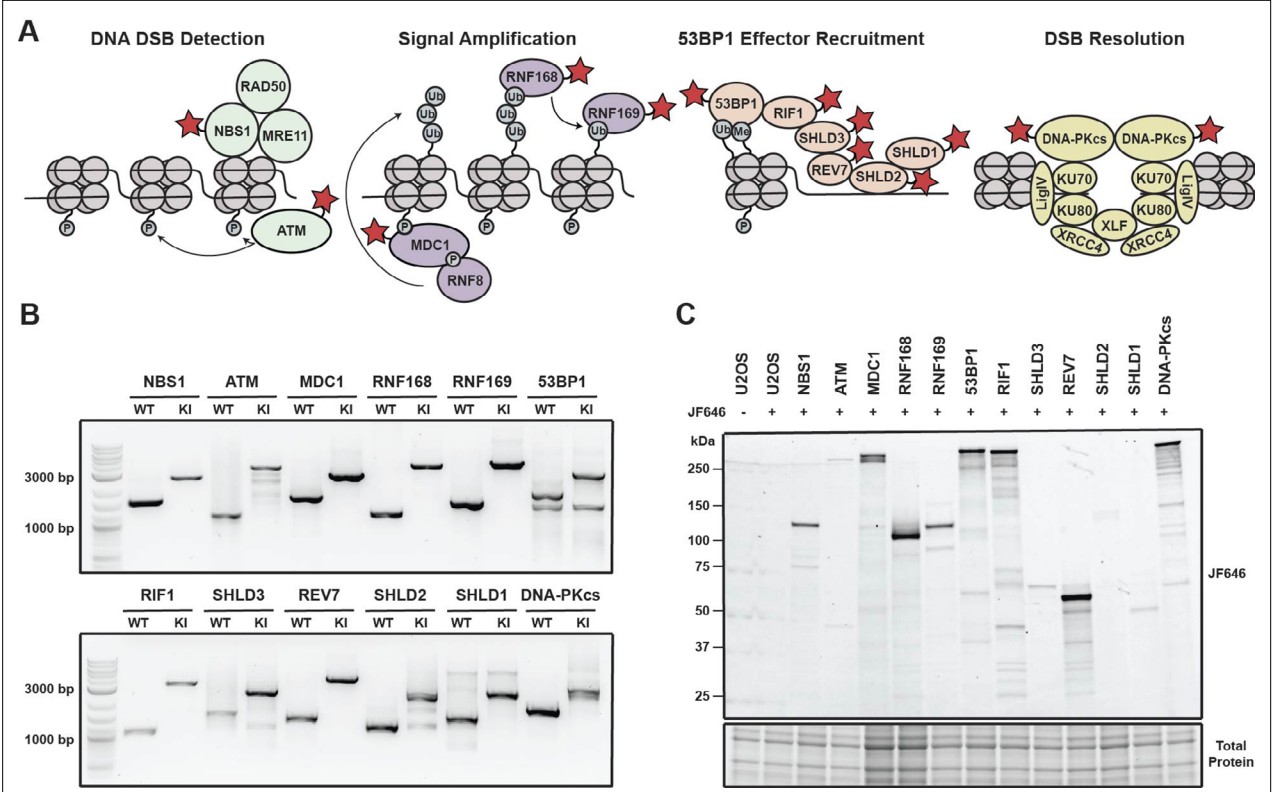

**Figure 1.** Generation of a panel HaloTagged DNA damage response proteins with CRISPR-Cas9 and homology-directed repair. (**A**) Model of the DDR factors HaloTagged by genome editing and their roles in DSB repair. Red star indicates HaloTagged protein. (**B**) Agarose gels depicting PCR products amplified from genomic DNA showing insertion of the 3xFLAG-HaloTag into the genomic loci of each tagged DDR factor using primers oriented outside of both left and right homology arms. (WT = wildtype; KI = knock in) (**C**) SDS-PAGE gel showing fluorescently labeled HaloTagged proteins in each cell line after labeling with JF646 HaloTag ligand.

The online version of this article includes the following source data and figure supplement(s) for figure 1:

**Figure supplement 1.** HaloTag knock-in design and knock-in validation.

**Source data 1.** Uncropped gels corresponding to *Figure 1*.

**Figure supplement 1—source data 1.** Uncropped gels and blots corresponding to *Figure 1—figure supplement 1*.

SHLD3, REV7, SHLD2, SHLD1, and DNA-PKcs). Using this panel of endogenously edited cell lines we systematically determined the absolute protein abundance of each protein, the kinetics of recruitment to laser-induced DSBs, and defined the diffusion dynamics and search mechanism of all factors using live-cell single-molecule imaging. We demonstrate that live cell single-molecule imaging is a highly sensitive method to detect the chromatin recruitment of DNA repair factors in response to DSB induction that is not limited to their accumulation in DNA repair foci. Furthermore, live-cell single-molecule imaging of the lowly abundant Shieldin complex components (SHLD1, SHLD2, SHLD3) demonstrates that the Shieldin complex does not exist as a preassembled complex but rather assembles at DNA lesions. Finally, live-cell single-molecule imaging reveals that MDC1 exists in a constitutive chromatin-associated state, which is mediated by MDC1's large unstructured PST repeat region and is independent of its BRCT domain. Altogether, our work provides new insight into the molecular mechanisms of DNA repair in human cells, establishes a new approach to analyze DNA repair factor recruitment to DNA lesions in living cells, and our panel of cell lines expressing HaloTagged DNA repair factors from their endogenous loci will be a powerful resource for the DNA repair field.

## Results

### Generation of a panel of endogenously HaloTagged DDR proteins by genome editing

To investigate the dynamics of DNA repair proteins at the single-molecule level in living cells, we used CRISPR-Cas9 and homology-directed repair to insert a 3x-FLAG-HaloTag at their endogenous genomic loci in U2OS cells (*Figure 1—figure supplement 1A*). U2OS cells have been widely used as a model cell line for DNA repair studies, however, it is important to note that U2OS cells do not express ATRX, a known contributing factor to HR (*Elbakry et al., 2021*; *Juhász et al., 2018*). We selected a variety of DNA repair factors that encompass various functional steps of DNA repair and the DNA damage response (DDR) including DNA double strand break (DSB) detection (ATM & NBS1), DDR signal amplification (MDC1 & RNF168), inhibition of DNA end resection (53BP1, RNF169, RIF1, SHLD3, REV7, SHLD2, and SHLD1), as well as DSB resolution (DNA-PKcs) (*Figure 1A*). Proteins were tagged at either the N-Terminus (MDC1, 53BP1, SHLD3, SHLD2, SDHL1, and DNA-PKcs) or the C-Terminus (NBS1, ATM, RNF168, RNF169, RIF1 & REV7) (*Figure 1—figure supplement 1B*).

We generated clonal cell lines for all targeted proteins and confirmed genome editing using genomic PCR and Sanger sequencing. All cell lines were homozygously edited except for Halo-SHLD2 which had one tagged allele and a frameshift mutation in the second allele (*Figure 1B* and *Figure 1—figure supplement 1C and E*). We obtained at least two clones for all knock-ins except for NBS1, 53BP1 and SHLD2 for which we obtained a single knock-in clone expressing only the HaloTagged protein. The presence of HaloTagged protein was validated by detection of fluorescently labeled proteins in an SDS-PAGE gel after labeling with the cell permeable HaloTag ligand JF646 (*Figure 1C*, *Figure 1—figure supplement 1D*). In addition, we confirmed that all cell lines exclusively expressed the tagged protein by western blotting when antibodies for these proteins were commercially available, which is critical to assess whether the HaloTagged proteins are fully functional (*Figure 1—figure supplement 1F*). HaloTagged proteins validated by Western blotting were expressed at or near endogenous protein levels (*Figure 1—figure supplement 1F*). Despite the commercial availability of antibodies to SHLD2, SHLD1, and RNF169, we were unsuccessful at detecting these proteins at endogenous expression levels. As an alternate approach for proteins where commercial antibodies were not available, we confirmed expression of these proteins by western blotting using an antibody directed to the 3 X FLAG epitope (RNF169, SHLD3, SHLD2, and SHLD1; *Figure 1—figure supplement 1D*). In total, we generated a panel of 12 clonal cell lines that exclusively express HaloTagged DDR factors from their endogenous loci.

### Functional validation of HaloTagged DDR proteins

To ensure that the HaloTagged DDR factors were functional, we assessed protein localization, recruitment to DNA damage induced foci, and clonogenic survival assays after treatment with the DSB-inducing drug Zeocin. To evaluate the subcellular localization of HaloTagged proteins, we imaged JF646-labeled HaloTagged proteins in live cells. As previously described, NBS1, MDC1, 53BP1, RIF1, RNF168, RNF169, and DNA-PKcs localized to the nucleus, whereas SHLD3, SHLD2, SHLD1, and REV7 were found in the cytoplasm and the nucleus (*Figure 2A*), suggesting that the HaloTag may not impact the proper cellular localization of these proteins (*Noordermeer et al., 2018*; *Wilson et al., 2016*). Most of these proteins also appeared to be excluded from nucleoli (*Figure 2A*). HaloTagging ATM at the N-terminus may lead to nuclear exclusion of the protein. To confirm that HaloTagging does not interfere with the recruitment of the tagged proteins to DNA damage sites, we analyzed their sub-cellular distribution by live-cell imaging one hour after inducing DNA DSBs with Zeocin, a radiomimetic drug chemically similar to bleomycin that also induces DNA single-strand breaks (*Povirk, 1996*; *Povirk et al., 1977*). While very few foci were detected in the absence of DNA damage in any of the cell lines, we observed dramatic increases in DNA damage induced foci formation for all proteins except for ATM and DNA-PKcs which would not be expected due to the limited number of these proteins that localize to DSBs (*Figure 2A*). This data confirms HaloTagged DDR proteins are capable of being recruited to DNA damage sites induced by Zeocin.

Finally, we tested sensitivity to Zeocin-induced DNA damage by clonogenic survival assays of all HaloTagged DDR clones compared to untagged parental U2OS cells. Considering the size of the HaloTag (34 kDa), it is possible that it could affect the biochemical activity of its fusion partner. If the HaloTag interfered with protein function in DNA repair, we expected to observe a decrease in

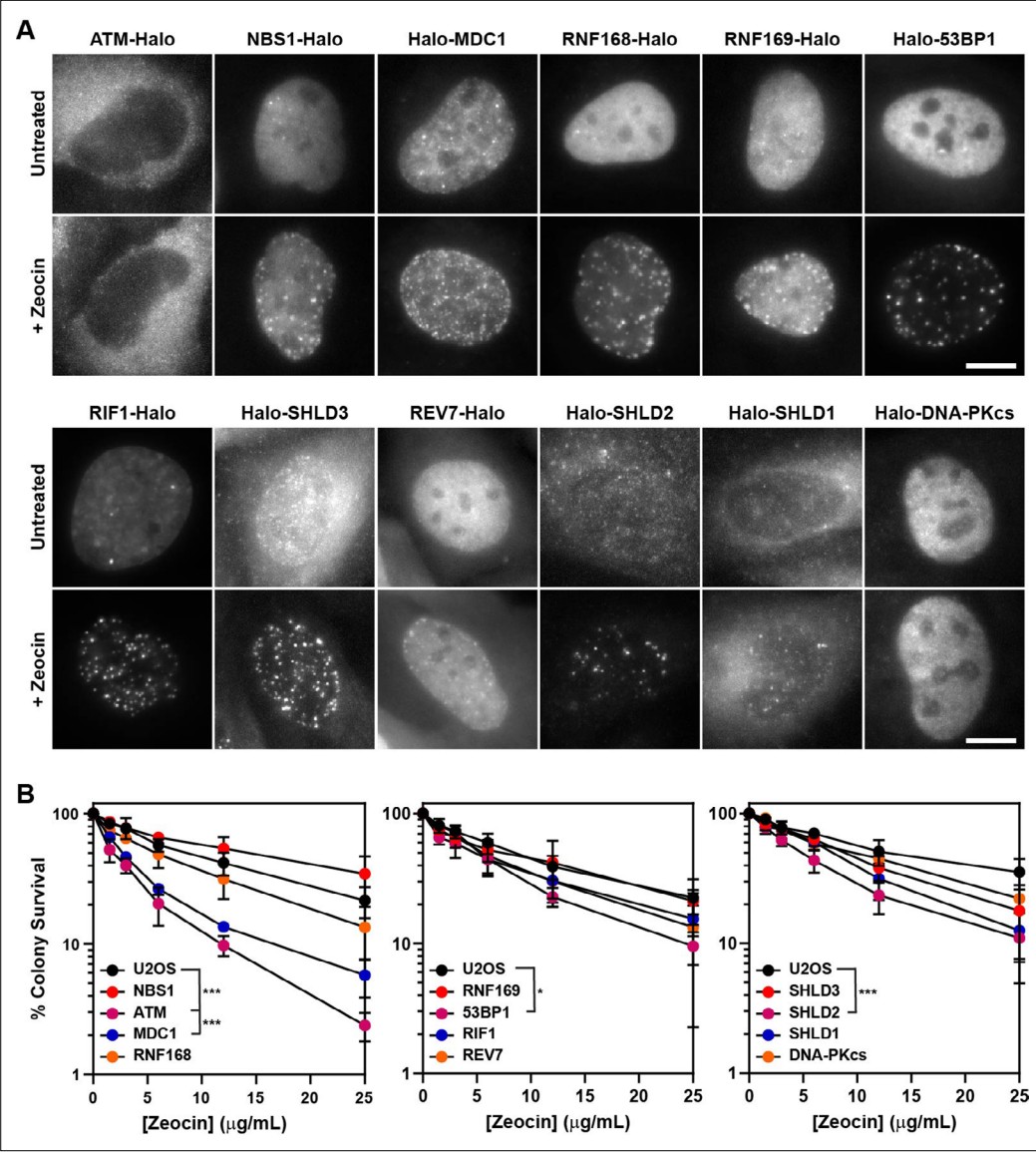

**Figure 2.** HaloTagged DDR proteins retain proper subcellular localization, foci-forming ability, and are competent for DNA repair. (**A**) Representative images of JF646-labeled HaloTagged proteins in the absence or presence of Zeocin in living cells. Data presented show protein cellular localization and foci-forming ability. Scale bar = 10 μm. Images are scaled differently between untreated and treated samples to demonstrate both localization and foci-forming ability. (**B**) Clonogenic survival assays representing the Zeocin-sensitivity of each HaloTagged DDR cell line relative to untagged parental U2OS cells. Data presented are the results of at least three independent experiments each plated in triplicate ± S.D. Data were compared by one-way ANOVA with Dunnett's posthoc test. * $p<0.05$; *** $p<0.001$.

The online version of this article includes the following source data and figure supplement(s) for figure 2:

**Figure supplement 1.** Functional validation of HaloTagged DDR proteins.

**Figure supplement 1—source data 1.** Uncropped gels and blots corresponding to *Figure 2—figure supplement 1*.

---

clonogenic survival after Zeocin treatment. For most of the HaloTagged proteins Zeocin sensitivity was indistinguishable from the parental U2OS cells, which was consistent between clones (*Figure 2B* and *Figure 2—figure supplement 1A*), confirming that these tagged proteins likely retain at least partial DNA repair function. Four cell lines were more sensitive to Zeocin than control cells including Halo-ATM, Halo-MDC1, Halo-53BP1, and Halo-SHLD2. The HaloTag-ATM cell line was as sensitive to

Zeocin as U2OS cells treated with the ATM inhibitor (ATMi) (KU-55933), which could not be alleviated by moving the HaloTag to the C-terminus of ATM, demonstrating that HaloTagging ATM at either end results in a non-functional protein (*Figure 2—figure supplement 1B–D*). To assess whether increased sensitivity to Zeocin induced DNA damage reflected a complete loss of protein function, we established 53BP1 and MDC1 knockout cell lines using CRISPR-Cas9. The Halo-53BP1 and Halo-MDC1 cell lines had modestly increased sensitivity to Zeocin but were significantly more resistant than their knockout counterparts, demonstrating they are partially functional (*Figure 2—figure supplement 1E–G*). Although the Halo-SHLD2 cell line is modestly sensitive to Zeocin in clonogenic assays, it does retain proper cellular localization consistent with observations from previous studies and is successfully recruited to DNA damage sites as demonstrated by its foci-forming ability (*Dev et al., 2018*; *Gupta et al., 2018*; *Noordermeer et al., 2018*). Additionally, we attempted to tag Halo-SHLD2 at the C-terminus but were unsuccessful in obtaining tagged clones. It is important to consider that HaloTag-induced changes in protein function could be critical variables in interpreting experimental results. While it is unclear for what reason HaloTagging has some functional impact on MDC1, 53BP1, and SHLD2, it appears to be independent of their ability to be recruited to DSB sites, as demonstrated by each protein's robust capacity to form foci in response to Zeocin-induced DNA damage. In summary, with the exception of ATM all of the HaloTagged proteins display expected subcellular localization within cells and are robustly recruited to DNA damage induced foci, and clonogenic assays suggest that most possess at least partial DNA repair functionality compared to their untagged counterparts.

## Quantification of absolute cellular protein abundance of DNA repair factors

A key determinant of the kinetics of DDR protein recruitment to DNA damage sites is the absolute cellular concentration of the respective DNA repair factor. For example, the core NHEJ factors, DNA-PKcs, Ku70, and Ku80, are highly abundant proteins which is thought to influence their rapid recruitment to DSBs (*Carter et al., 1990*; *Cho et al., 2022*; *Mimori et al., 1986*). Additionally, very little is known about the absolute abundance of members of the Shieldin complex in cells which are thought to be some of the least abundant proteins in the human proteome (*Gupta et al., 2018*). Thus, determining the absolute cellular protein abundances is critical to establish a mechanistic, quantitative model of the DNA damage response. To measure the protein abundance of the tagged DNA repair factors, we used in-gel fluorescence and flow cytometry-based methodologies. For in-gel fluorescence quantification of the HaloTagged proteins, we modified a method originally described by *Cattoglio et al., 2019*. To perform these experiments, we generated a standard curve using known quantities of recombinant 3XFLAG-HaloTag labeled with JF646 and cell lysates from a specific number of U2OS cells (*Figure 3A*). Two independent clones of each cell line were labeled with 500 nM JF646, which was ~10 x higher than the saturating concentration for the most abundant protein, REV7 (*Figure 3—figure supplement 1A*). Because each protein migrates differently on SDS-PAGE gels based on its size and amino acid composition, we considered the possibility that this may influence the fluorescence of each sample relative to the purified 3XFLAG-HaloTag. To account for differences in the fluorescence signal cause by different migration rates and patterns we cleaved the HaloTag from each protein prior to SDS-PAGE using TEV protease to create an adjustment factor for each protein (*Figure 3—figure supplement 1B*). We obtained a TEV correction factor for each protein except RNF168, RNF169, and MDC1 which all appeared to be degraded after cell lysis which could not be alleviated by including a protease inhibitor cocktail (*Figure 3—figure supplement 1C*). Correction factors ranged 1.04 for SHLD1 to 3.04 for SHLD2 meaning that cleaved HaloTag fluorescence intensity was 1.04x – 3.04x greater for the cleaved HaloTag, than the full-length fusion protein (*Figure 3—figure supplement 1D*).

After correcting for differences in fluorescence intensity cause by SDS-PAGE migration patterns, HaloTagged protein abundance ranged from ~1600 (ATM) – 180,000 (REV7) molecules per cell (*Table 1*), with ATM (~1600–4200 molecules per cell), SHLD1 (~7300–8100 molecules per cell), SHLD2 (~7300 molecules per cell), and SHLD3 (~4000–5600 molecules per cell) clones having the lowest expression level (*Table 1*). Considering a large proportion of SHLD1, SHLD2, and SHLD3 protein is localized to the cytoplasm, the nuclear protein abundance for each of these proteins is even lower which may influence the kinetics of Shieldin complex recruitment in 53BP1-dependent NHEJ. We observed higher protein abundances for factors involved in initial break detection

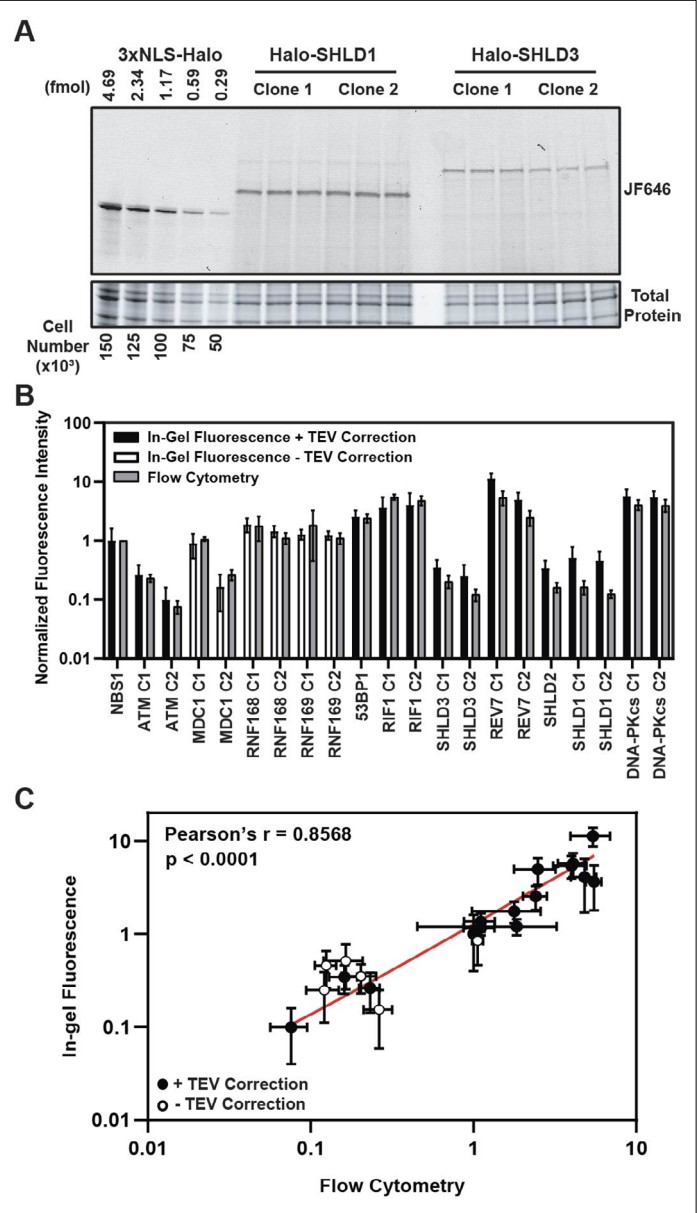

**Figure 3.** HaloTag enables quantification of absolute cellular protein abundances. (**A**) Example image of in-gel fluorescence of JF646-labeled HaloTagged proteins. (**B**) Comparison of JF646 fluorescence intensity values (normalized to NBS1-Halo) between in-gel fluorescence after applying the TEV correction factor and flow cytometry. White columns indicate unadjusted samples because of the inability to accurately determine a TEV correction factor due to protein degradation. Data are presented as the mean of ≥ three independent experiments ± S.D. (**C**) Plot representing the correlation of normalized JF646 fluorescence intensities ± S.D. for each protein between TEV-corrected in-gel fluorescence and flow-cytometry. Data were analyzed by Pearson's correlation coefficient. White points indicate those proteins for which a TEV-correction factor could not be accurately determined. Red line represents an interpolated standard curve.

The online version of this article includes the following source data and figure supplement(s) for figure 3:

**Figure supplement 1.** Quantification of absolute protein abundance using HaloTagged DDR proteins.

**Source data 1.** Uncropped gels corresponding to *Figure 3*.

**Figure supplement 1—source data 1.** Uncropped gels corresponding to *Figure 3—figure supplement 1*.

**Table 1.** Absolute protein abundances of HaloTagged DNA repair proteins in U2OS cells.
Left column: Absolute protein abundance of HaloTagged proteins determined by in-gel fluorescence after adjusting with the TEV correction factor calculated for each protein. An asterisk (*) indicates the three proteins for which an accurate TEV correction factor could not be generated. For TEV-corrected samples, the S.D. includes propagated error. Right column: Determination of absolute protein abundances of untagged DNA repair proteins in parental U2OS cells by applying a Western blot correction factor to adjust for differences in expression between HaloTagged and untagged protein. For western blot corrected samples, S.D. includes propagated error. An asterisk (*) denotes samples where the Western correction was applied to samples that were not TEV corrected.
No Data indicates the absence of a commercially available antibody or an antibody that detects endogenous levels of protein expression.

| Protein | HaloTag Protein ± S.D. | Western Correction ± S.D. |
|---|---|---|
| NBS1 | 15,846±9547 | 18,055±11,180 |
| ATM C1 | 4161±1897 | 4738±2379 |
| ATM C2 | 1578±945 | 3115±2365 |
| MDC1 C1 | 13,352±6001* | 16,030±7631* |
| MDC1 C2 | 2434±1506 | 3390±2180* |
| RNF168 C1 | 27,787±7241* | 4585±1819* |
| RNF168 C2 | 21,531±4801* | 4678±1145 |
| RNF169 C1 | 19,013±3759* | No Data |
| RNF169 C2 | 18,413±3119* | No Data |
| 53 BP1 | 40,162±11,498 | 12,109±5792 |
| RIF1 C1 | 57,486±28,970 | 9382±5765 |
| RIF1 C2 | 64,358±37,504 | 10,398±7506 |
| SHLD3 C1 | 5555±1913 | No Data |
| SHLD3 C2 | 3953±2187 | No Data |
| REV7 C1 | 178,131±40,757 | 39,803±10,299 |
| REV7 C2 | 78,340±25,161 | 66,790±42,830 |
| SHLD2 | 7257±3104 | No Data |
| SHLD1 C1 | 8124±4104 | No Data |
| SHLD1 C2 | 7257±3103 | No Data |
| DNA-PKcs C1 | 90,899±26,030 | 115,143±46,933 |
| DNA-PKcs C2 | 85,786±23,508 | 138,704±71,488 |

including NBS1 (~16,000 molecules per cell) and DNA-PKcs (~86,000–91,000 molecules per cell) and those contributing to DDR signal amplification including MDC1 (~2400–13,000 molecules per cell), RNF168 (~22,000–28,000 molecules per cell), and RNF169 (~18,000–19,000 molecules per cell) (*Table 1*). The highest protein abundances were observed for 53BP1 (~40,000 molecules per cell), RIF1 (~58,000–64,000 molecules per cell), REV7 (~78,000–180,000 molecules per cell), and DNA-PKcs (~86,000–91,000 molecules per cell) (*Table 1*). Importantly, the differences in absolute protein number between independent genome edited clones could be the consequence of either a different number of alleles being modified with the HaloTag, considering the possibility of inherent heterogeneity in karyotype within U2OS cells, or alternatively could be due to variations in protein expression within different clones. As a complementary approach, we used flow cytometry to measure the fluorescence intensity of JF646-labeled HaloTagged proteins in cells (*Figure 3—figure supplement 1E*). The protein abundance values we obtained by in-gel fluorescence were compared to flow-cytometric quantifications of mean fluorescence intensity for each cell line. Flow-cytometry does not provide an

absolute protein number, but instead measures relative protein abundance between the cell lines expressing different HaloTagged proteins. The relative fluorescence levels detected by flow cytometry corresponded well with the relative absolute protein abundance determined by in-gel fluorescence (*Figure 3B and C*). Furthermore, results with this complementary approach were comparable to in-gel fluorescence for proteins where a TEV correction factor could not be accurately determined indicating that a TEV correction would have only led to modest changes in overall abundance of these proteins.

Finally, using western blotting for each protein where an antibody was commercially available and could detect endogenous levels of protein expression, we quantified the concentration of each HaloTagged protein relative to untagged wild-type protein in parental U2OS cells. Expression of HaloTagged proteins relative to untagged protein in U2OS cells ranged from 0.62 x and 0.79 x for HaloTagged DNA-PKcs clones to 6.13 x and 6.19 x for RIF1 clones with 53BP1, RNF168, and one REV7 clone also being expressed at higher levels upon HaloTagging, potentially as a consequence of increased protein stabilization or increased protein production (*Figure 3—figure supplement 1F*). We used the relative expression of each protein compared to untagged protein to calculate the number of molecules per cell for each protein in parental U2OS cells (*Table 1*). After applying this adjustment factor, DNA-PKcs was the most abundant protein ranging from ~71,000 to 120,000 molecules per cell. Conversely, ATM and RNF168 had the lowest protein abundances ranging from ~3100 to 4700 molecules per cell. Additionally, the calculated number of molecules per cell for both clones expressing each tagged protein fell within the standard deviation of each other with the exception of MDC1. This high variability of calculated molecules per cell between MDC1 clones is likely attributable to inconsistent signals in western blots which was not alleviated by optimizing multiple wet transfer conditions or by using two separate MDC1 antibodies. In summary, in-gel fluorescence enabled quantification of absolute protein abundance of HaloTagged proteins in U2OS cells with a dynamic range capable of detecting both lowly and highly expressed proteins.

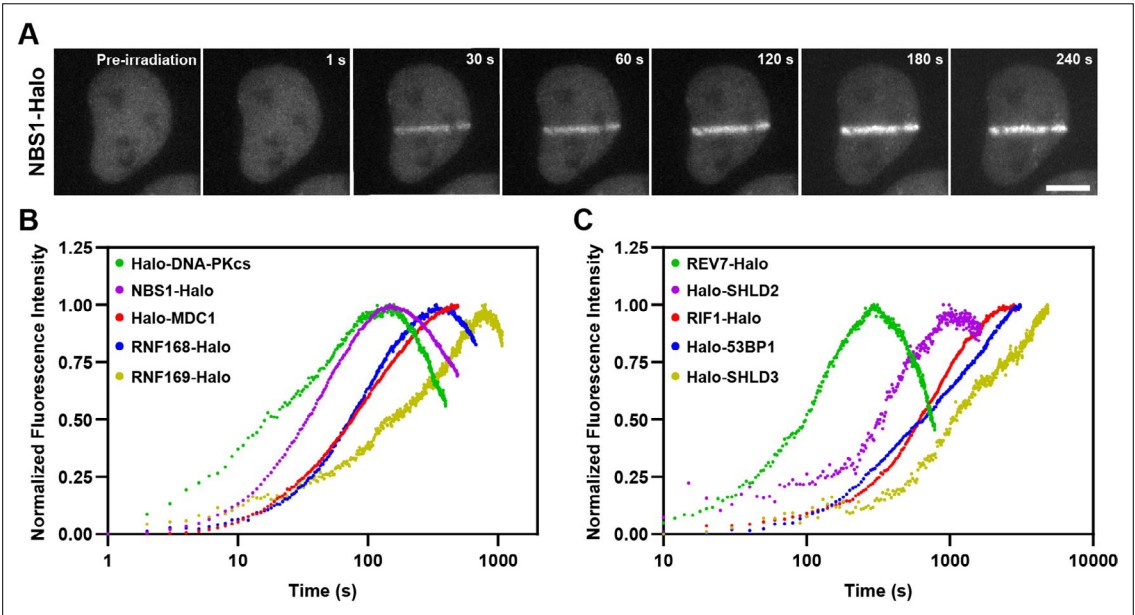

**Figure 4.** Kinetics of HaloTagged DDR proteins recruitment to sites of laser microirradiation induced DNA breaks. (**A**) Representative images of NBS1-Halo (JFX650) recruitment to laser-induced DSBs over time (Scale bar = 10 μm). (**B**) Normalized recruitment kinetics of HaloTagged DNA-PKcs, NBS1, MDC1, RNF168, and RNF169 proteins to laser-induced DSBs. (**C**) Normalized recruitment kinetics of HaloTagged 53BP1, RIF1, REV7, SHLD2, and SHLD3 proteins to laser-induced DSBs. Data are presented as the average increase in fluorescence post-laser microirradiation normalized to the brightest average frame for each movie. n=8–13 individual cells analyzed for each HaloTag cell line.

The online version of this article includes the following figure supplement(s) for figure 4:

**Figure supplement 1.** Kinetics of HaloTagged DDR protein recruitment to laser-induced DSBs.

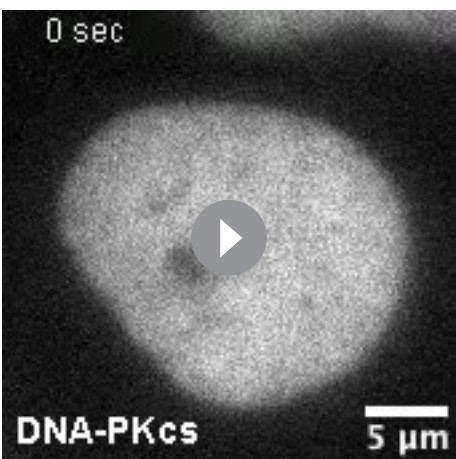

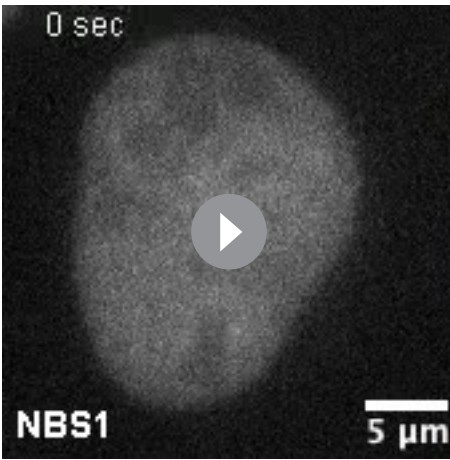

**Video 1.** Representative movie demonstrating recruitment of JFX650-labeled 3xFLAG-HaloTagged DNA-PKcs to DNA DSBs after laser microirradiation. Images were acquired at one frame per second. 170x170 pixels with a pixel size of 0.16 µm.
https://elifesciences.org/articles/87086/figures#video1

**Video 2.** Representative movie demonstrating recruitment of JFX650-labeled 3xFLAG-HaloTagged NBS1 to DNA DSBs after laser microirradiation. Images were acquired at one frame per second. 170x170 pixels with a pixel size of 0.16 µm.
https://elifesciences.org/articles/87086/figures#video2

## Kinetics of recruitment of HaloTag DDR proteins to sites of laser-induced microirradiation

After validating that our panel of HaloTagged DDR factors are functional and proficient in DNA repair, we monitored the kinetics of recruitment for each factor to sites of laser-induced DNA damage. While laser-induced microirradiation (LMI) induces DNA DSBs, it also causes a variety of other types of DNA damage including single-strand breaks and base damage and likely induces DNA damage at levels typically not encountered by human cells (*Holton et al., 2017*; *Muster et al., 2017*). Even considering these important drawbacks, LMI has been widely used as one tool to monitor recruitment kinetics of DNA repair proteins to spatially and temporally controlled sites of DNA damage. Most previous studies that monitored the kinetics of recruitment of DNA repair factors to laser-induced DNA damage sites have used transgene expression of fluorescently tagged proteins, which could influence their recruitment kinetics due to changes in protein levels and competition with the untagged endogenous protein. The use of HaloTagged proteins expressed from their endogenous loci allowed us to more accurately determine the relative recruitment kinetics of each DNA repair factor to laser-induced DNA lesions. HaloTagged DNA repair factors were labeled with JFX650 and cells were pre-sensitized to LMI with Hoechst (1 µg/mL). LMI reproducibly induced robust protein recruitment for each DNA repair factor (*Figure 4A*, *Figure 4—figure supplement 1A*, *Videos 1–10*). To compare the relative recruitment kinetics of the DNA repair factors we determined the time to half-maximal accumulation ($t_{1/2}$) (*Figure 4—figure supplement 1B*). DNA-PKcs and NBS1 accumulated rapidly after LMI, with a $t_{1/2}$ = 22.1 s and $t_{1/2}$ = 31.3 s, respectively, consistent with their known roles in the early steps of the DNA damage response (*Figure 4B* and *Figure 4—figure supplement 1B*). MDC1 ($t_{1/2}$ = 76.5 s) and RNF168 ($t_{1/2}$ = 69.4 s) arrived at similar times, consistent with rapid

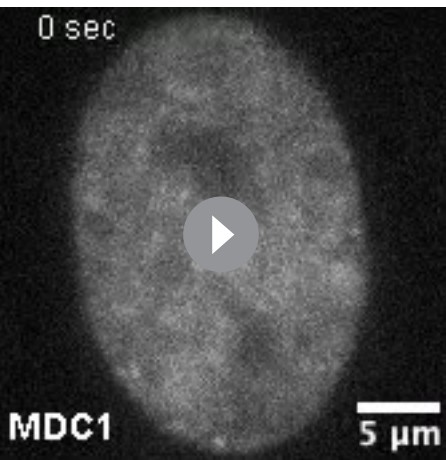

**Video 3.** Representative movie demonstrating recruitment of JFX650-labeled 3xFLAG-HaloTagged MDC1 to DNA DSBs after laser microirradiation. Images were acquired at one frame per second. 170x170 pixels with a pixel size of 0.16 µm.
https://elifesciences.org/articles/87086/figures#video3

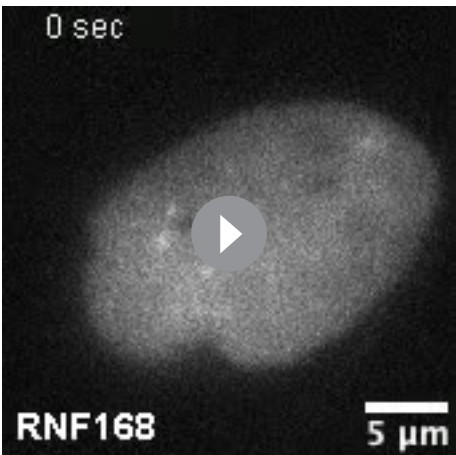

**Video 4.** Representative movie demonstrating recruitment of JFX650-labeled 3xFLAG-HaloTagged RNF168 to DNA DSBs after laser microirradiation. Images were acquired at one frame per second. 170x170 pixels with a pixel size of 0.16 µm.
https://elifesciences.org/articles/87086/figures#video4

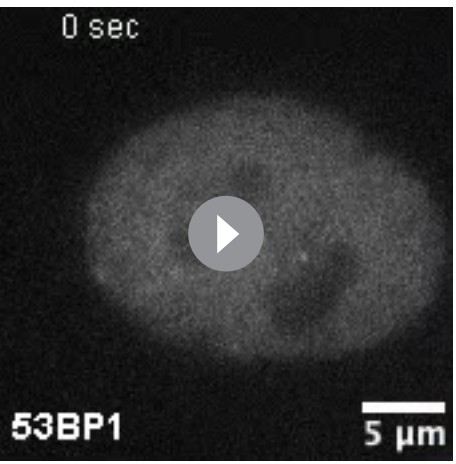

**Video 6.** Representative movie demonstrating recruitment of JFX650-labeled 3xFLAG-HaloTagged 53BP1 to DNA DSBs after laser microirradiation. Images were acquired at one frame every 10 s. 170x170 pixels with a pixel size of 0.16 µm.
https://elifesciences.org/articles/87086/figures#video6

ubiquitin deposition at DSBs dependent upon MDC1 mediated RNF8 and RNF168 recruitment (*Figure 4B* and *Figure 4—figure supplement 1B*). RNF169 ($t_{1/2}$ = 185.8 s) was significantly delayed compared to RNF168 (*Figure 4B* and *Figure 4—figure supplement 1B*), indicating that RNF169 requires chromatin modification by RNF168 prior to its recruitment. 53BP1 and RIF1 both accumulated slowly at LMI sites with comparable kinetics ($t_{1/2}$ = 669.0 s, $t_{1/2}$ = 550.8 s, respectively) (*Figure 4C* and *Figure 4—figure supplement 1B*). REV7 ($t_{1/2}$ = 90.8 s) and SHLD3 ($t_{1/2}$ = ~1168 s), which are thought to form a complex reached their half-maximal accumulation at LMI induced DNA lesions with distinct kinetics, although REV7 possesses additional functions in DNA repair beyond the Shieldin complex, which could lead to the observed recruitment kinetics but was not further analyzed in this study (*Figure 4C* and *Figure 4—figure supplement 1B*). Importantly, we did not analyze recruitment kinetics of REV7-Halo in the absence of SHLD3. In addition, SHLD2 ($t_{1/2}$ = 287.1 s) is recruited more rapidly than 53BP1 and RIF1, which suggests that either

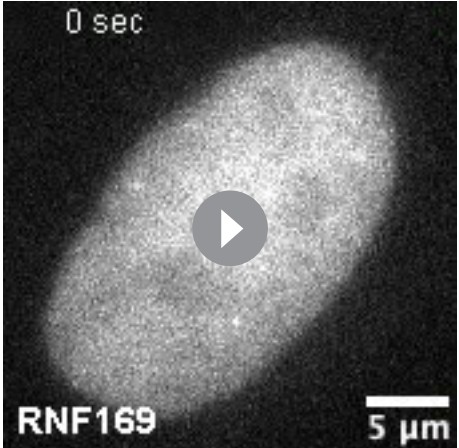

**Video 5.** Representative movie demonstrating recruitment of JFX650-labeled 3xFLAG-HaloTagged RNF169 to DNA DSBs after laser microirradiation. Images were acquired at one frame per second. 170x170 pixels with a pixel size of 0.16 µm.
https://elifesciences.org/articles/87086/figures#video5

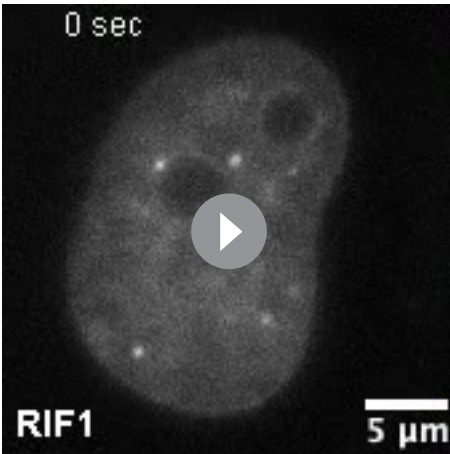

**Video 7.** Representative movie demonstrating recruitment of JFX650-labeled 3xFLAG-HaloTagged RIF1 to DNA DSBs after laser microirradiation. Images were acquired at one frame every 10 s. 170x170 pixels with a pixel size of 0.16 µm.
https://elifesciences.org/articles/87086/figures#video7

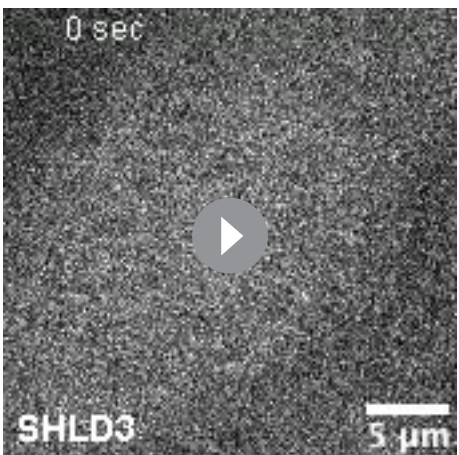

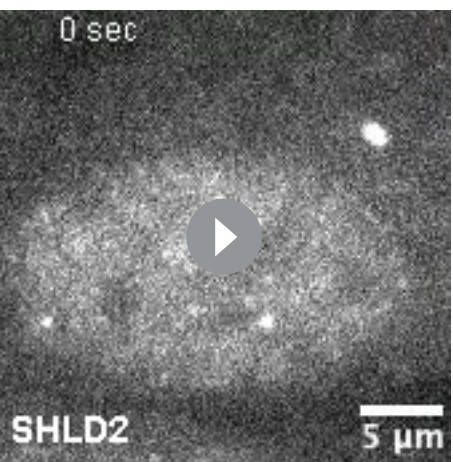

**Video 8.** Representative movie demonstrating recruitment of JFX650-labeled 3xFLAG-HaloTagged SHLD3 to DNA DSBs after laser microirradiation. Images were acquired at one frame every 10 s. 170x170 pixels with a pixel size of 0.16 μm.

https://elifesciences.org/articles/87086/figures#video8

**Video 10.** Representative movie demonstrating recruitment of JFX650-labeled 3xFLAG-HaloTagged SHLD2 to DNA DSBs after laser microirradiation. Images were acquired at one frame every five seconds. 170x170 pixels with a pixel size of 0.16 μm.

https://elifesciences.org/articles/87086/figures#video10

SHLD2 is not strictly dependent on these proteins and can be recruited to DSBs by its association with ssDNA, or that at the time SHLD2 begins to accumulate at DSB sites there is already sufficient 53BP1, RIF1, and SHLD3 for SHLD2 recruitment. To distinguish between these possibilities, we knocked out SHLD3 in Halo-SHLD2 cells and imaged SHLD2 in the presence or absence of Zeocin-induced DSBs in live-cells (*Figure 4—figure supplement 1C*). While SHLD3 wildtype cells supported SHLD2 foci-formation after Zeocin treatment, loss of SHLD3 completely eliminated SHLD2 foci-formation, consistent with the well-known genetic dependency of SHLD2 on SHLD3 and with the hypothesis that at the time SHLD2 accumulation begins sufficient SHLD3 is present at DNA breaks to initiate SHLD2 recruitment. To confirm this by comparing the absolute amount of Halo-SHLD2 and Halo-SHLD3 recruited to laser induced DNA damage sites, we repeated the experiments using identical imaging conditions and without intensity normalization. In these experiments Halo-SHLD2 and Halo-SHLD3 were simultaneously recruited to DNA lesions in comparable amounts (*Figure 4—figure supplement 1D*). After Halo-SHLD2 accumulation plateaued, Halo-SHLD3 continued to accumulate in excess of Halo-SHLD2 (*Figure 4—figure supplement 1D*). Finally, we also observed recruitment of SHLD1 to laser-induced DSBs but were limited to acquiring images every ten minutes due to technical issues with photo-bleaching of the fluorescence signal at faster imaging rates that resulted from low SHLD1 expression and slow recruitment to LMI sites (*Figure 4—figure supplement 1E*). Interestingly, SHLD1 is recruited in very low amounts and had the appearance of single particles enriched in the area of laser microirradiation, but the signal intensity was insufficient to reliably determine the recruitment kinetics of SHLD1. The recruitment kinetics of Halo-RNF168 and Halo-RNF169 are comparable to those observed in a previous study that systematically analyzed DNA repair factor recruitment to laser induced damage sites (*Aleksandrov et al., 2018*). The $t_{1/2}$ data for Halo-MDC1 and Halo-53BP1 we observed

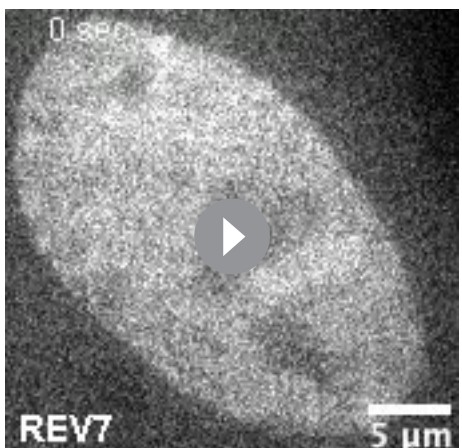

**Video 9.** Representative movie demonstrating recruitment of JFX650-labeled 3xFLAG-HaloTagged REV7 to DNA DSBs after laser microirradiation. Images were acquired at one frame every two seconds. 170x170 pixels with a pixel size of 0.16 μm.

https://elifesciences.org/articles/87086/figures#video9

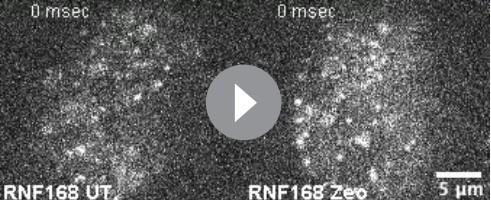

**Video 11.** Representative live-cell single-molecule imaging movies of untreated and zeocin-treated U2OS cells expressing 3xFLAG-HaloTagged NBS1 labeled with JFX650 and acquired at 138 frames per second. 170x140 pixels with a pixel size of 0.16 µm.
https://elifesciences.org/articles/87086/figures#video11

**Video 13.** Representative live-cell single-molecule imaging movies of untreated and zeocin-treated U2OS cells expressing 3xFLAG-HaloTagged RNF168 labeled with JFX650 and acquired at 138 frames per second. 170x140 pixels with a pixel size of 0.16 µm.
https://elifesciences.org/articles/87086/figures#video13

differed by approximately 2-fold from those reported by *Aleksandrov et al., 2018*. These differences could be a consequence of the expression method (Bacmid transgene expression vs. genome editing) or the species from which the DNA repair factor was derived (mouse 53BP1 vs. human 53BP1). In summary, we have determined the recruitment kinetics of the HaloTagged DNA repair factors expressed at endogenous or near endogenous expression levels. The wide range of observed kinetics provides insights into the relative recruitment of the analyzed DDR factors to DSBs and the stoichiometry of Shieldin complex subunits.

## Single-molecule live-cell imaging of HaloTagged DDR proteins

The diffusion dynamics of DNA repair factors in the nucleus are a key determinant of their rapid and specific recruitment to DNA lesions. Live-cell single-molecule imaging makes it possible to analyze the diffusion dynamics, chromatin binding, and recruitment to sites of DNA damage of individual DNA repair proteins in their endogenous context. To carry out single-molecule imaging of the HaloTagged DNA repair factors, we used a combination of sparse HaloTag labeling and highly inclined laminated optical sheet (HILO) microscopy (*Tokunaga et al., 2008*), and imaged cells at 138 frames per second to capture rapid diffusion dynamics in both unperturbed conditions and after treatment with Zeocin (*Videos 11–21*). Single particles were automatically detected in each frame, linked into trajectories using the multi-target tracking algorithm (*Sergé et al., 2008*), and step size distributions were analyzed using the Spot-On tool to extract the diffusion coefficients of freely diffusing ($D_{\text{free}}$) and chromatin bound molecules ($D_{\text{bound}}$), as well as the fraction of particles that are associated with chromatin ($F_{\text{bound}}$) (*Figure 5A*; *Hansen et al., 2018*). As model proteins for freely diffusing and chromatin bound factors, we transiently expressed and imaged the 3XFLAG-HaloTag fused to a 3 x Nuclear Localization Sequence (3XNLS) and HaloTagged histone H2B, respectively (*Figure 5—figure supplement 1A*, *Video 22*). 3XFLAG-HaloTag-3XNLS ($D_{\text{free}}$ = 3.9 µm²/s, $F_{\text{bound}}$ = 16%) represents the lower bound for static particles and the upper bound for the free diffusion coefficient, while Halo-H2B ($F_{\text{bound}}$ = 66%) represents the upper bound for the fraction of bound particles since histone H2B is an integral component of chromatin. The fraction of bound molecules ($F_{\text{bound}}$) values for Halo-NLS and Halo-H2B were

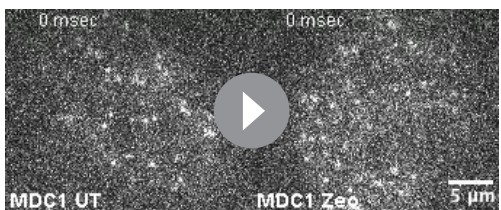

**Video 12.** Representative live-cell single-molecule imaging movies of untreated and zeocin-treated U2OS cells expressing 3xFLAG-HaloTagged MDC1 labeled with JFX650 and acquired at 138 frames per second. 170x140 pixels with a pixel size of 0.16 µm.
https://elifesciences.org/articles/87086/figures#video12

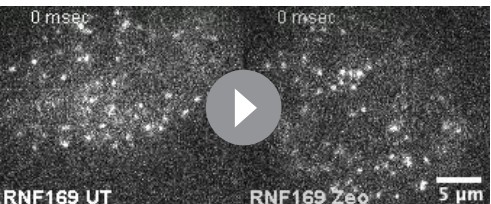

**Video 14.** Representative live-cell single-molecule imaging movies of untreated and zeocin-treated U2OS cells expressing 3xFLAG-HaloTagged RNF169 labeled with JFX650 and acquired at 138 frames per second. 170x140 pixels with a pixel size of 0.16 µm.
https://elifesciences.org/articles/87086/figures#video14

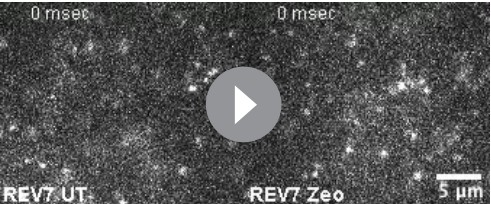

**Video 15.** Representative live-cell single-molecule imaging movies of untreated and zeocin-treated U2OS cells expressing 3xFLAG-HaloTagged 53BP1 labeled with JFX650 and acquired at 138 frames per second. 170x140 pixels with a pixel size of 0.16 μm.
https://elifesciences.org/articles/87086/figures#video15

**Video 17.** Representative live-cell single-molecule imaging movies of untreated and zeocin-treated U2OS cells expressing 3xFLAG-HaloTagged REV7 labeled with JFX650 and acquired at 138 frames per second. 170x140 pixels with a pixel size of 0.16 μm.
https://elifesciences.org/articles/87086/figures#video17

similar to those previously reported in *Hansen et al., 2018*, which did not include a 3XFLAG tag fused to the HaloTag. Conversely, the free diffusion coefficients ($D_{free}$) values we report were lower than previously reported, due to the well described impact of the higher labeling density used in our experiments on diffusion coefficient measurements of highly dynamic proteins (*Hansen et al., 2018*). For all proteins, we only analyzed nuclear particle trajectories and assumed a two-state diffusion model where particles either freely diffuse or are chromatin bound (*Figure 5—figure supplement 1B*). Importantly, determining $D_{free}$, $D_{bound}$, and $F_{bound}$ for individual cells (*Figure 5A–C*, *Figure 5—figure supplement 1C*), and combining the step size measurements from all cells for three experimental replicates (*Figure 5—figure supplement 1D*) lead to similar results, indicating that our analysis approach is robust. The diffusion coefficient for freely moving particles ranged from $D_{free}$ = 1.0–3.7 μm²/s, which are all lower than the diffusion coefficient measured for the HaloTag-3XNLS (*Figure 5B*). The diffusion coefficients measured for SHLD1 ($D_{free}$ = 3.7), SHLD2 ($D_{free}$ = 1.8), and SHLD3 ($D_{free}$ = 2.3) were significantly different from each other. This suggests that the shieldin complex is not preassembled in the nucleoplasm, which is consistent with the distinct recruitment kinetics to LMI induced sites of DNA damage we observed for the shieldin complex components (*Figure 4C*). Furthermore, intensity profiles of MDC1 and RIF1 trajectories provided confidence that the vast majority of particle localizations represent single fluorophores and not multiple labeled molecules co-diffusing or intersecting which would have additive effects on intensity values (*Figure 5—figure supplement 1E*).

The diffusion coefficient for bound particles in unperturbed cells ranged from $D_{bound}$ = 0.02–0.10 μm²/s, which are all faster than Halo-H2B (*Figure 5C*). One would predict to observe differences in the $D_{bound}$ between factors that are physically incorporated within nucleosomes (e.g. H2B), bind modified histone tails (e.g. MDC1), or accumulate into phase separated compartments (e.g. 53BP1). Consistent with this expectation, Halo-H2B had a $D_{bound}$ of 0.012 μm²/s, while the diffusion coefficient for bound MDC1 molecules was two-fold higher ($D_{bound}$ = 0.024 μm²/s). On the other hand, 53BP1 which can diffuse within phase-separated compartments had the fastest diffusing bound particles ($D_{bound}$ = 0.099 μm²/s). These data point to the power of this method to detect even small differences in the

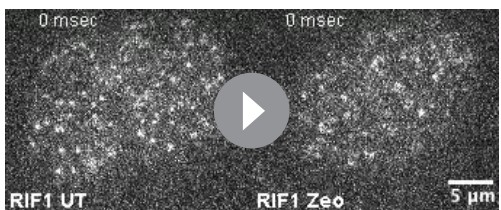

**Video 16.** Representative live-cell single-molecule imaging movies of untreated and zeocin-treated U2OS cells expressing 3xFLAG-HaloTagged RIF1 labeled with JFX650 and acquired at 138 frames per second. 170x140 pixels with a pixel size of 0.16 μm.
https://elifesciences.org/articles/87086/figures#video16

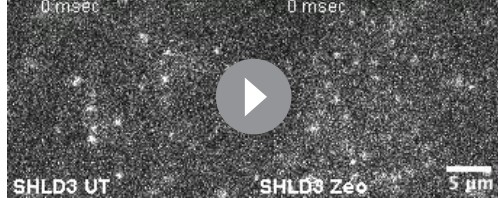

**Video 18.** Representative live-cell single-molecule imaging movies of untreated and zeocin-treated U2OS cells expressing 3xFLAG-HaloTagged SHLD3 labeled with JFX650 and acquired at 138 frames per second. 170x140 pixels with a pixel size of 0.16 μm.
https://elifesciences.org/articles/87086/figures#video18

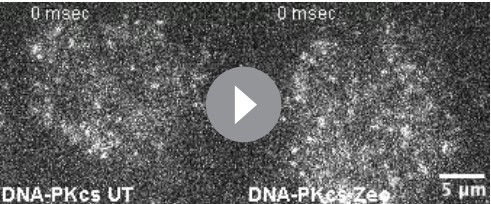

**Video 19.** Representative live-cell single-molecule imaging movies of untreated and zeocin-treated U2OS cells expressing 3xFLAG-HaloTagged SHLD2 labeled with JFX650 and acquired at 138 frames per second. 170x140 pixels with a pixel size of 0.16 μm.
https://elifesciences.org/articles/87086/figures#video19

**Video 21.** Representative live-cell single-molecule imaging movies of untreated and zeocin-treated U2OS cells expressing 3xFLAG-HaloTagged DNA-PKcs labeled with JFX650 and acquired at 138 frames per second. 170x140 pixels with a pixel size of 0.16 μm.
https://elifesciences.org/articles/87086/figures#video21

behavior of DNA repair factors when bound to chromatin. We observed a wide range of the fraction of molecules bound to chromatin ($F_{bound}$ = 23–69%) in unperturbed cells (*Figure 5D*). Importantly, the chromatin bound fraction of all DNA repair factors analyzed was significantly higher than that of HaloTag-3XNLS, which likely reflects their intrinsic propensity to associate with chromatin (*Figure 5D*). Alternatively, the chromatin bound molecules could be the result of low levels of DNA lesions present in control cells. Strikingly, the chromatin bound fraction of MDC1 ($F_{bound}$ = 69%) and RIF1 ($F_{bound}$ = 68%) in unperturbed cells was comparable to that of HaloTag-H2B ($F_{bound}$ = 66%), suggesting that these DNA repair factors are constitutively associated with chromatin and may search for DNA breaks by local chromatin scanning (*Figure 5D*).

After Zeocin treatment, we observed a significant increase in the fraction of static particles for RNF169, 53BP1, SHLD2, SHLD3, and DNA-PKcs (*Figure 5D*), consistent with their recruitment to DNA breaks. Since MDC1 and RIF1 were largely chromatin associated in control cells it is unsurprising that we did not observe a further increase in their overall chromatin association after zeocin treatment (*Figure 5D*). It is important to note that a change in the fraction bound requires that a significant percentage of the respective DNA repair factor has to be recruited to DNA lesions, which depends on the abundance of the repair factor relative to the number of DSBs induced. NBS1, REV7, and RNF168 are among the most abundant proteins analyzed, which could explain why their $F_{bound}$ did not significantly increase after zeocin treatment (*Figure 5D*). Additionally, HaloTagging RNF168 leads to expression ~4–6 x that of wild-type protein in U2OS cells which could also contribute to our inability to detect significant changes in protein binding after Zeocin treatment. Despite its low abundance SHLD1 we were not able to detect recruitment of SHLD1 to chromatin after DSB induction using this approach (*Figure 5C*), consistent with the very limited and slow recruitment we observed in the LMI experiments described above (*Figure 4—figure supplement 1D*). Importantly, for all factors that showed an increase in the $F_{bound}$ after zeocin treatment, we observed a decrease in the $D_{free}$ (*Figure 5B and D*), which is likely the result of a systematic error as a consequence of the global step size fitting in Spot-On.

While these observations report on global changes in DDR protein behavior, they did not distinguish

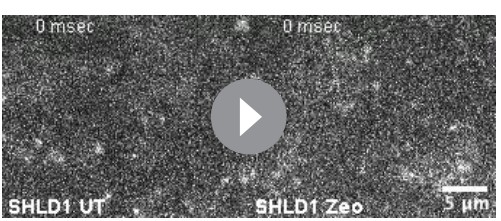

**Video 20.** Representative live-cell single-molecule imaging movies of untreated and zeocin-treated U2OS cells expressing 3xFLAG-HaloTagged SHLD1 labeled with JFX650 and acquired at 138 frames per second. 170x140 pixels with a pixel size of 0.16 μm.
https://elifesciences.org/articles/87086/figures#video20

between how particles behave within or outside of repair foci, which have been traditionally used as a marker of DNA repair factor recruitment to DNA lesions. By combining sparse labeling of Halo-53BP1 by JFX650 (pseudo-colored green) with subsequent quantitative labeling using the spectrally distinct JF503 (pseudo-colored magenta) we could simultaneously image single 53BP1 molecules and DNA repair foci marked by 53BP1 accumulation at Zeocin-induced DSBs (*Figure 5—figure supplement 1F*; *Videos 23 and 24*). We observed highly static and dynamic 53BP1 particles within 53BP1 foci consistent with the relatively high $D_{bound}$ we observed for 53BP1

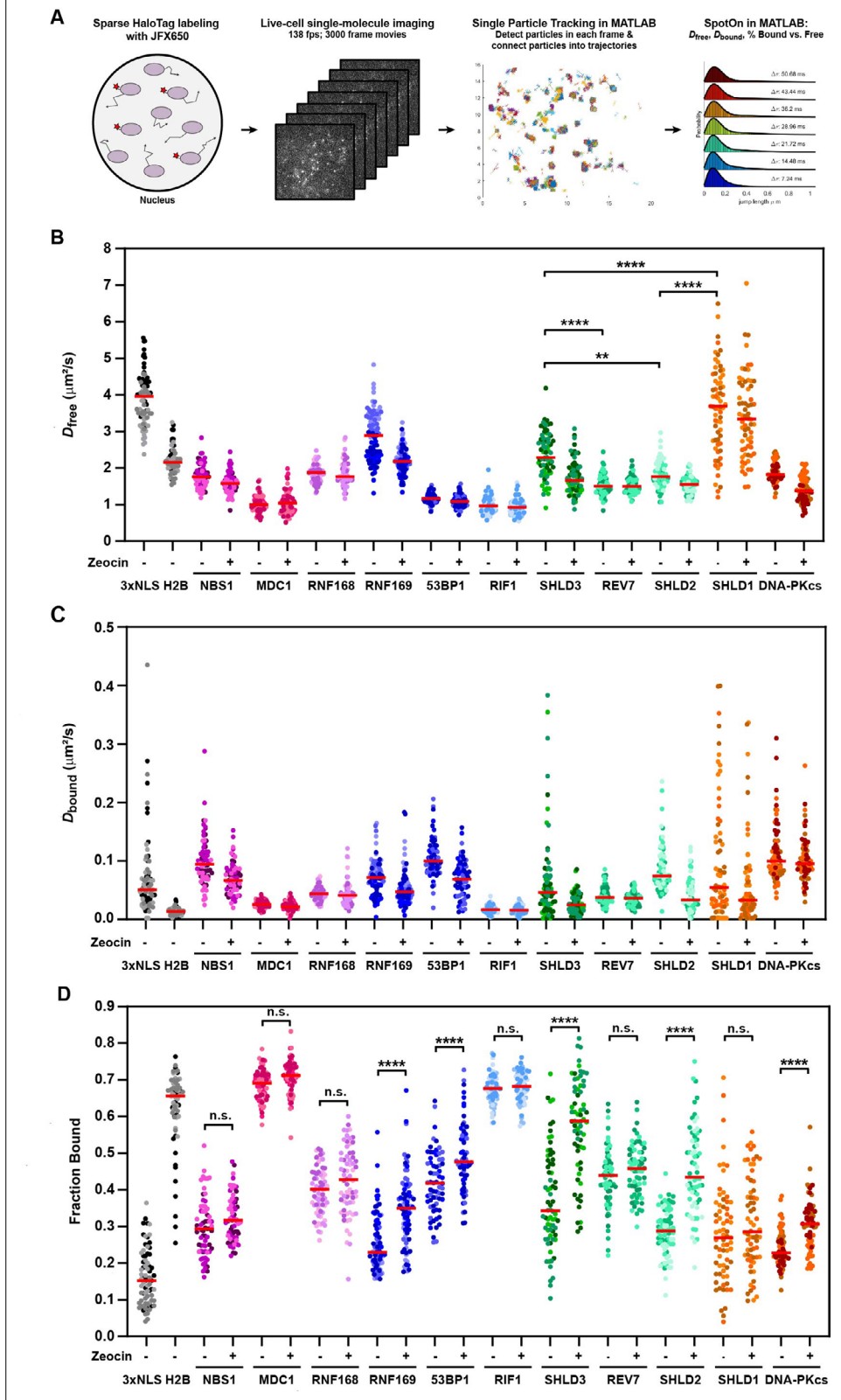

**Figure 5.** HaloTagged DDR proteins exhibit distinct nuclear diffusion and chromatin binding characteristics. (**A**) Graphical representation of the workflow used for live-cell single-molecule imaging of HaloTagged DDR proteins. (**B**) Diffusion coefficients for freely diffusing HaloTag DDR proteins present in at least three consecutive frames in untreated conditions and post-Zeocin exposure. Values plotted indicate the $D_{free}$ for all analyzed tracks per cell

*Figure 5 continued on next page*

*Figure 5 continued*

with each dot indicating a separate cell that was analyzed. Live-cell single-molecule imaging was performed over 3–4 separate days imaging at least 20 individual cells per condition per experimental replicate (n≥60 cells total for each protein and condition representing three to four independent experiments). Red bar = median. (**C**) Diffusion coefficients of the bound fraction of HaloTag DDR proteins present in at least three consecutive frames in untreated conditions and post-Zeocin exposure. Values plotted indicate the $D_{bound}$ for all analyzed tracks per cell with each dot indicating a separate cell that was analyzed. Live-cell single-molecule imaging was performed over 3–4 separate days imaging at least 20 individual cells per condition per experimental replicate (n≥60 cells total for each protein and condition). Red bar = median. (**D**) Plot of the Fraction Bound for each HaloTag DDR protein under each condition that were analyzed using a two-state model of diffusion. Each dot represents the fraction bound of each protein in an individual cell (n≥60 cells for each protein and condition). Red bar = median. Differently shaded points indicate data collected from separate biological experiments. Data were analyzed by two-way ANOVA with Tukey's posthoc test. n.s.=not significant. ** p=0.004; **** p<0.0001.

The online version of this article includes the following source data and figure supplement(s) for figure 5:

**Figure supplement 1.** Nuclear diffusion and chromatin binding characteristics of HaloTagged DDR proteins.

**Figure supplement 1—source data 1.** Uncropped gels corresponding to *Figure 5—figure supplement 1*.

---

(*Figure 5—figure supplement 1G*, *Videos 23–24*). Static 53BP1 molecules within DNA repair foci likely represent molecules directly bound to chromatin while dynamic 53BP1 molecules are likely recruited by the formation of phase-separated 53BP1 compartments (*Figure 5—figure supplement 1G*, *Videos 23–24*), as observed by others (*Kilic et al., 2019*). We also observed static 53BP1 particles not associated with DNA repair foci which could represent the initial recruitment of 53BP1 to chromatin prior to the formation of a phase-separated compartment or DNA lesions that are rapidly repaired and do not mature into a detectable DNA repair focus (*Figure 5—figure supplement 1G*,

*Videos 23–24*). In addition, we detected the transition of freely diffusing 53BP1 molecules in and out of DNA repair foci (*Figure 5—figure supplement 1G*, *Videos 23–24*). To further dissect the dynamics of Halo-53BP1 within and outside of DNA repair foci, we used a mask of DNA repair foci to split Halo-53BP1 trajectories into tracks that overlapped with DNA repair foci and those that did not (*Figure 5—figure supplement 1F*). The trajectories were then analyzed with Spot-On using a three-state model assuming Halo-53BP1 molecules could either be freely diffusing, directly bound to chromatin, or part of a phase-separated DNA repair focus, which would lead to an intermediate diffusion coefficient. Within DNA repair foci 29% of 53BP1 molecules moved with a slow diffusion coefficient ($D_{free2}$=0.6 μm²/s) compared

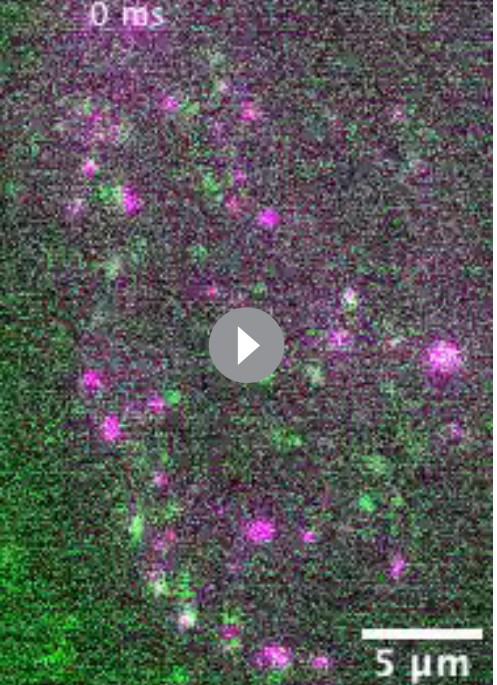

**Video 23.** Representative live-cell single-molecule imaging movie of Zeocin-treated U2OS cells expressing endogenous 3xFLAG-HaloTag-53BP1 sparsely labeled with JFX650 (pseudo-colored green) acquired at 202 frames per second and overlayed with a Z-projected image of 53BP1 foci which were densely labeling with JF503 (pseudo-colored magenta). Pixel size = 0.108 μm.
https://elifesciences.org/articles/87086/figures#video23

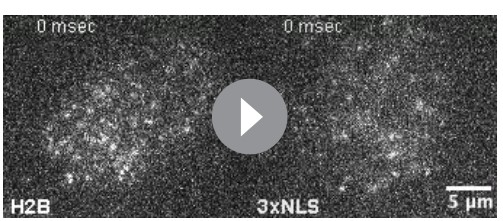

**Video 22.** Representative live-cell single-molecule imaging movies of U2OS cells transiently expressing HaloTag-H2B and HaloTag-3xNLS labeled with JFX650 and acquired at 138 frames per second. 170x140 pixels with a pixel size of 0.16 μm.
https://elifesciences.org/articles/87086/figures#video22

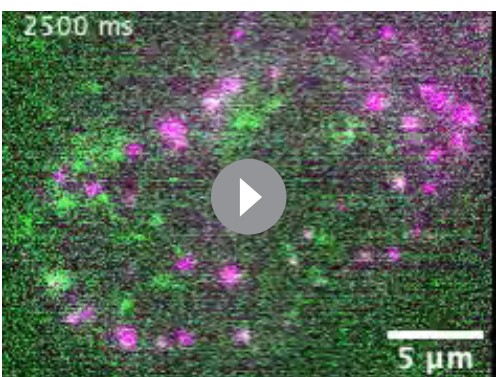

**Video 24.** A second representative live-cell single-molecule imaging movie of Zeocin-treated U2OS cells expressing endogenous 3xFLAG-HaloTag-53BP1 sparsely labeled with JFX650 (pseudo-colored green) acquired at 202 frames per second and overlayed with a Z-projected image of 53BP1 foci which were densely labeling with JF503 (pseudo-colored magenta). Pixel size = 0.108 μm.

https://elifesciences.org/articles/87086/figures#video24

to 17% of 53BP1 trajectories that did not overlap with DNA repair foci (*Figure 5—figure supplement 1H*). This observation is consistent with the presence of two types of 53BP1 molecules within DNA repair foci, one that is directly associated with chromatin and a second that is potentially recruited via a phase-separation mechanism. Taken together, combined imaging of single-particles and DNA repair foci could be a powerful tool enabling analyses to specifically monitor or differentiate between protein behaviors inside and outside of repair foci.

In summary, live-cell single-molecule imaging provides insight into the molecular mechanism by which DNA repair factors search for DNA lesions, reports on complex formation, and allows the analysis of their recruitment to DNA lesions. In addition, our observations demonstrate that DNA repair factor recruitment to chromatin is not strictly confined to DNA repair foci and therefore could report on DNA repair events that were previously not detectable.

## Residence time analysis of chromatin-bound DNA repair factors

To gain further insight into the biochemical properties of the DNA repair factors bound to chromatin, we analyzed the residence time of static single-particle trajectories. Accurate residence time analysis requires continuous tracking of bound molecules without any gaps in the trajectories. To robustly track long-lasting binding events, we averaged the intensity values of 10 consecutive imaging frames, which amplifies the signal of static molecules and reduces the intensity of mobile particles because their signal is spread out over multiple pixels over the course of 10 time points (*Figure 6A*). Single-particles were then tracked by constraining the diffusion coefficient to only detect chromatin bound molecules (*Video 25*). Importantly, when analyzing long-lived single-molecule fluorescence signals it is critical determine the contribution of photo-bleaching to signal disappearance. To assess the photo-bleaching rate under our imaging conditions, we analyzed the integrated the nuclear signal intensity of cells expressing Halo-H2B over time, which limits the contribution of protein dissociation and diffusion of fluorescent molecules out of the focal plane to the loss of fluorescence signal over time. The fluorescence half-life of Halo-H2B was 7.7 seconds under our imaging conditions (*Figure 6B*), which sets the upper limit for our residence time analysis. We used an aggregate data set including residence times from three biological replicates (>60 cells per cell line), since individual replicates lead to highly reproducible results (*Figure 6—figure supplement 1A*). The residence time for all factors analyzed, including Halo-H2B, fit well to two exponential decay functions (*Figure 6B*, *Figure 6—figure supplement 1B*), indicating all factors exhibit transient and stable binding to chromatin. Importantly, the half-life of stable single Halo-H2B molecules binding to chromatin matched the half-life observed of total nuclear fluorescence, indicating that our measurement of the photo-bleaching rate is accurate and H2B residence time exceeds the 22 s time frame of our experiment (*Figure 6B*). All DNA repair factors analyzed dissociated from chromatin more rapidly than histone H2B with half-lives ranging from $t_{1/2}$ = 2.4 seconds for Halo-53BP1 to $t_{1/2}$ = 5.8 s for Halo-MDC1 (*Figure 6C*). The induction of DNA damage using zeocin increased to residence time of Halo-53BP ($t_{1/2\ ZEO}$ = 3.6 s), Halo-SHLD2 ($t_{1/2\ ZEO}$ = 4.9 s), and Halo-SHLD3 ($t_{1/2\ ZEO}$ = 5.8 s) (*Figure 6C*), consistent with the increase in the chromatin bound fraction of these proteins observed in our Spot-On analysis (*Figure 5D*). In contrast, the residence time of the remaining factors was unchanged in the presence of zeocin, which suggests that the biochemical interactions that trigger their chromatin association are comparable under both experimental conditions. Importantly for all factors analyzed a fraction of molecules were present for the entire imaging time (MDC1 $F_{>22s}$=3%, 53BP1 $F_{>22s}$=0.2%, *Figure 6B*). We failed to assess the

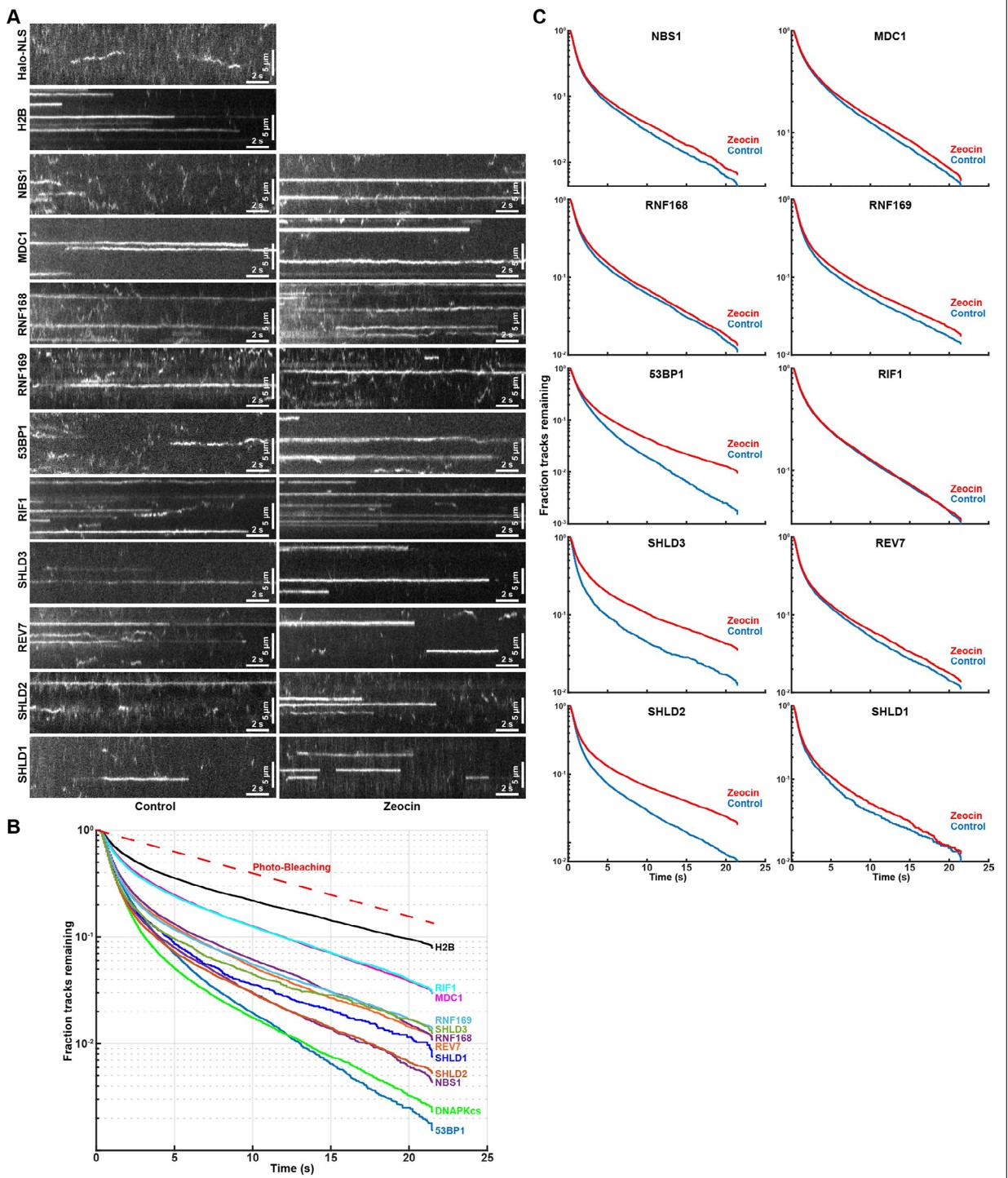

**Figure 6.** Residence time analysis of DNA repair factors in response to DNA damage. (**A**) Kymographs of single-molecule imaging movies for HaloTagged DNA repair factors. Movies were acquired with 7.2ms exposure times and to amplify long lasting interactions the intensity of 10 consecutive frames was averaged. (**B**) Residence time (track length) distribution of long-lasting chromatin binding events of DNA repair factors displayed as survival probability after the elapsed time. (**C**) Residence time (track length) distribution of long-lasting chromatin binding for the indicated DNA repair factors displayed as survival probability after the elapsed time. Aggregated data from 3 to 4 separate days imaging at least 20 individual cells per condition per experimental replicate (n≥60 cells total for each protein and condition) was fit with two exponential decay functions (R$^2$ >0.99 for all proteins) and half-lives and fractions of slow and fast decaying components are reported.

The online version of this article includes the following figure supplement(s) for figure 6:

**Figure supplement 1.** Residence time analysis of DNA repair factors in response to DNA damage.

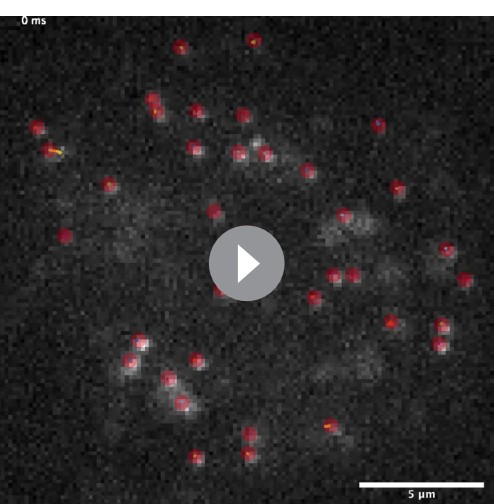

**Video 25.** Representative movie demonstrating robust tracking of bound particles after averaging the image intensity over 10 frames and constraining the expected diffusion coefficient of bound particles to $D_{max}$ = 0.05 $\mu m^2/s$. Images acquired at 138 frames per second and the frame rate was reduced to 13.8 frames per second by the averaging over 10 consecutive frames. Pixel size = 0.16 $\mu m$.

https://elifesciences.org/articles/87086/figures#video25

residence time of Halo-DNA-PKcs because the high concentration of rapidly diffusing Halo-DNA-PKcs molecules lead to high background signal, making it impossible to reliably track statically bound Halo-DNA-PKcs molecules. In total, these results demonstrate that DNA break induction by zeocin and the associated changes in chromatin state (histone modifications, DNA break resection) increase the affinity of the association of 53BP1, SHLD2, and SHLD3 with DNA breaks. It is important to note that due to photo-bleaching and potential drift of the bound molecules out of the focal plane of the objective these experiments underestimate the absolute time of chromatin binding of the proteins analyzed.

## Live-cell single-molecule imaging reveals MDC1's constitutive chromatin interaction is mediated by its PST repeat domain

Our live-cell single-molecule imaging revealed that MDC1 is constitutively chromatin associated. MDC1 contains two domains that have been implicated in chromatin binding: The BRCT domains that bind to γH2AX (*Stucki et al., 2005*), and its 13 PST repeats, which have been proposed to associate with the nucleosome acidic patch (*Salguero et al., 2019*). To dissect the relative contribution of the BRCT and PST repeat domains to the constitutive chromatin association of MDC1, we transiently expressed HaloTagged Halo-MDC1 wildtype (WT), PST deletion (ΔPST), or BRCT deletion (ΔBRCT) mutants in MDC1 knockout (ΔMDC1) cells (*Figure 7A–B*). We first assessed the ability of these MDC1 variants to localize to DNA damage induced foci. WT MDC1, and MDC1 ΔPST formed foci after zeocin treatment, while MDC1 ΔBRCT was not recruited to DNA damage induced foci (*Figure 7C*), consistent with previous observations that BRCT-γH2AX binding is essential for MDC1 foci formation and does not require the PST domain (*Stucki et al., 2005*). This suggests that the BRCT domains of MDC1 are the primary driver of MDC1 recruitment to DNA lesions. Next, we performed live-cell single-molecule imaging to define the contribution of the BRCT and PST repeat domains to the constitutive chromatin association we observed in unperturbed cells (*Figure 7D*, *Figure 7—figure supplement 1A–B*, *Video 26*). WT MDC1 transiently expressed in ΔMDC1 behaved identically to endogenously tagged MDC1 (*Figure 7D*, *Figure 7—figure supplement 1A–B*, *Figure 5B–C*), and deletion of the BRCT domains did not alter the diffusion behavior of MDC1 (*Figure 7D*, *Figure 7—figure supplement 1A–B*). In contrast, deletion of the PST repeat domain almost completely eliminated the fraction of MDC1 molecules associated with chromatin (*Figure 7D*, *Figure 7—figure supplement 1A–B*). This suggests that the PST repeat domain of MDC1 mediates the constitutive chromatin association of MDC1 even in the absence of DNA damage. Because a high fraction of MDC1 particles are chromatin-bound in unperturbed cells, we suspected that this interaction masks the ability to detect γH2AX-bound MDC1 after Zeocin exposure in single-molecule imaging experiments (*Figure 5D*). Therefore, we analyzed the single-molecule dynamics of the MDC1 deletion mutants in the presence or absence of Zeocin (*Figure 7—figure supplement 1C*). Consistent with our earlier observations, no changes in the bound fraction of MDC1 were detected after Zeocin treatment in cells expressing full-length MDC1 or the BRCT deletion mutant (*Figure 7—figure supplement 1C*). In contrast, upon deletion of the PST repeat region, we observed a significant increase in chromatin-bound MDC1 after Zeocin exposure (*Figure 7—figure supplement 1C*). This observation supports the conclusion that the PST-mediated constitutive interaction of MDC1 with chromatin masked the ability to detect BRCT-domain mediated binding of MDC1 to γH2AX at DNA breaks by live-cell single-molecule imaging. To analyze the contribution of the PST- and BRCT-domains

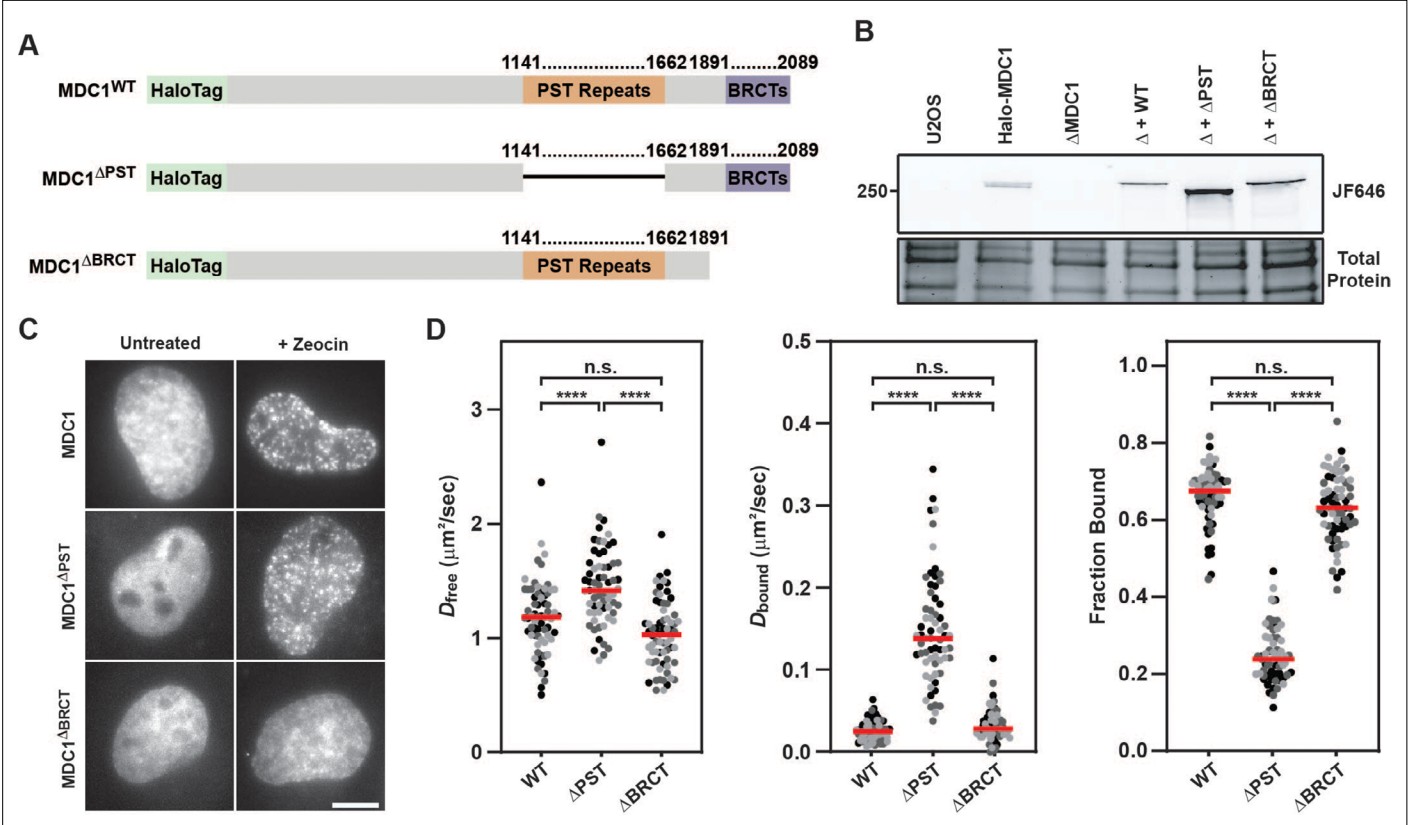

**Figure 7.** MDC1's constitutive chromatin association is mediated by its PST repeat region. (**A**) Graphical illustration of the primary sequence of MDC1 indicating the location of the PST repeat and BRCT domain and the associated deletion mutants generated to analyze effects on the MDC1-chromatin interaction. (**B**) SDS-PAGE gel of JF646-labeled cells depicting expression of transiently expressed WT, ΔPST, and ΔBRCT MDC1 in Halo-MDC1 knockout cells. (**C**) Representative images of transiently expressed, JF646-labeled MDC1 deletion mutants in living cells in the presence of absence of Zeocin. (**D**) Results of live-cell single-molecule analysis of transiently overexpressed MDC1 deletion mutants analyzed with single particle tracking and SpotOn. Each dot represents the indicated $D_{free}$, $D_{bound}$, or Fraction Bound for MDC1 molecules appearing in at least three consecutive frames within a single cell. Red bar = median. Data are the combination of all analyzed cells imaged over three independent experiments (n≥60 cells total). Differently shaded points indicate data collected from separate biological replicates. Data were analyzed by two-way ANOVA with Tukey's posthoc test. n.s.=not significant. ****=p < 0.0001.

The online version of this article includes the following source data and figure supplement(s) for figure 7:

**Figure supplement 1.** MDC1's constitutive chromatin association is mediated by its PST repeat region.

**Source data 1.** Uncropped gels corresponding to *Figure 7*.

---

to DSB repair, we used CRISPR-Cas9 to knock-in the DR-GFP HR reporter into the *AAVS1* locus in MDC1 knockout cells which we confirmed by genomic DNA PCR (*Figure 7—figure supplement 1D*; *Pierce et al., 1999*). The DR-GFP HR reporter uses a double stranded break induced in a non-functional GFP gene by the restriction enzyme I-SceI to assess repair by HR through reconstituting a functional

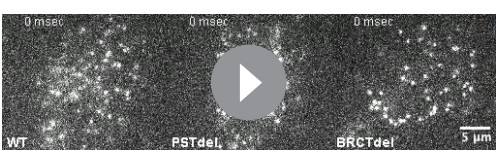

**Video 26.** Representative live-cell single-molecule imaging movies of HaloTagged MDC1 deletion mutants transiently expressed in ΔMDC1 U2OS cells, labeled with JFX650, and acquired at 138 frames per second. 170x140 pixels with a pixel size of 0.16 µm.
https://elifesciences.org/articles/87086/figures#video26

GFP gene by a repair donor contained in the reporter cassette that lacks a functional promotor (*Pierce et al., 1999*). We transiently transfected HaloTagged MDC1 variants and the I-SceI expression plasmid into the DR-GFP reporter cells in which endogenous MDC1 was knocked out and used flow cytometry to detect GFP expressing cells which resulted from HR mediated repair of the GFP reporter. Expression of HaloTagged WT MDC1 led to a significantly higher number of GFP positive cells compared to MDC1 knockout cells,

consistent with HaloTagged MDC1 being capable of supporting HR (*Figure 7—figure supplement 1E*). In contrast, HR efficiency was significantly reduced to a similar degree when MDC1 lacking the PST- or BRCT-domains was expressed, which was previously documented by others (*Xie et al., 2007*). Additionally, we used immunofluorescence to determine the ability of HaloTagged MDC1 variants to support recruitment of 53BP1 to Zeocin-induced DSBs. Compared to untreated cells, WT and ΔPST MDC1 were robustly recruited into DNA repair foci, while inhibiting γH2AX binding by deleting the BRCT domain prevented MDC1 foci formation consistent with previous observations (*Figure 7—figure supplement 1F*; *Xie et al., 2007*). Additionally, HaloTagged WT and ΔPST MDC1 were sufficient to promote recruitment of 53BP1 to DSB sites, while 53BP1 recruitment was defective in ΔMDC1 cells or cells expressing MDC1 lacking the BRCT domain (*Figure 7—figure supplement 1F*). Taken together, these observations support a model in which MDC1 is constitutively tethered to chromatin by its PST repeat region and is enriched at DNA lesions via the interaction of the BRCT domains with γH2AX. Importantly, both the PST- and BRCT-domains are critical for efficient HR while the PST domain is not required for 53BP1 recruitment to DSBs, suggesting that while MDC1-chromatin binding is important for HR, this interaction may be dispensable for 53BP1-dependent DNA end-joining.

## Discussion

In this study, we have developed a panel of cell lines expressing HaloTagged DNA repair factors from their endogenous genomic loci. Using these cell lines, we have systematically analyzed the protein abundance, diffusion dynamics, and recruitment to DNA lesions of these factors, which are critical to maintain genome integrity in human cells. Our observations provide new insights into the molecular mechanism and kinetics of the recruitment of the shieldin complex to DNA lesions, which is a critical step in DSB repair pathway choice. In addition, our results reveal how the PST repeat and BRCT domains of MDC1 coordinate its chromatin binding and recruitment to DNA lesions to facilitate DSB repair. In total, our work is an important step towards developing a quantitative model of DNA repair in human cells and provides a number of new tools which will be tremendously useful for the DNA repair research community.

### The shieldin complex is recruited to DNA lesions in distinct steps

The shieldin complex is essential to facilitate NHEJ downstream of 53BP1 by recruiting the CST complex and Polα/primase to DSBs (*Mirman et al., 2018*). While the pairwise protein and genetic interactions of the shieldin components are well understood (*Gupta et al., 2018*), how their dynamic recruitment to DSBs regulates repair via NHEJ was unknown. For instance, it is unclear to what extent shieldin is preassembled either as a complex (SHLD3-REV7-SHLD2-SHLD1), as subcomplexes (e.g. SHLD3-REV7 and SHLD2-SHLD1), or assembled entirely at a DSB. Some evidence supports the hypothesis of assembly at a break. Genetically SHLD3 is upstream of REV7, which in turn recruits SHLD2. SHLD2 directly associates with both single-stranded DNA and SHLD1, which recruits the CST complex to promote end fill-in (*Gupta et al., 2018*; *Mirman et al., 2022*; *Noordermeer et al., 2018*). Our live-cell single-molecule imaging demonstrates that SHLD1, SHLD2, and SHLD3 move through the nucleus with distinct diffusion coefficients. The diffusion coefficient of REV7 is comparable to SHLD2, but distinct from SHLD3. Observations by others have demonstrated that the interaction between SHLD3 and REV7 forms with extraordinarily slow kinetics which could be a key regulatory step in shieldin assembly at sites of DNA damage (*Susvirkar and Faesen, 2022*). Together our data strongly suggests that the shieldin complex components do not exist in a preassembled state, but rather associate at DNA lesions, which could allow the interaction of SHLD3 with REV7 to be rate limiting for shieldin complex formation.

Quantification of cellular protein abundance for HaloTagged SHLD1, SHLD2, and SHLD3 revealed that these proteins are expressed at comparable low levels. In addition, we observed both cytoplasmic and nuclear localization of each of these proteins by live-cell imaging. Therefore, the nuclear concentration for SHLD1, SHLD2, and SHLD3 is even lower. The low abundance and nuclear exclusion of SHLD1, 2, and 3 suggests that the recruitment of these factors to DSBs is tightly controlled. REV7 is largely found in the nucleus and 10-fold more abundant than SHLD3, which could significantly accelerate their slow complex formation observed in vitro (*Susvirkar and Faesen, 2022*). The low abundance of these proteins also has implications for understanding how these proteins find sites

of DNA damage. For example, does recruitment of SHLD1, 2, and 3 occur by freely diffusing molecules encountering DSBs or could enrichment at DSBs be promoted by the establishment of 53BP1-mediated phase separated repair compartments (*Kilic et al., 2019*)? The low abundance of SHLD2 also raises the question of how it can compete for ssDNA binding with RPA which is thought to be more abundant and has a high affinity for ssDNA (*Cho et al., 2022*; *Kim et al., 1992*). One explanation could be the local enrichment of SHLD2 at DSBs mediated by 53BP1, comparable to POT1 recruitment to telomeric ssDNA overhangs by the shelterin complex (reviewed in *Litman Flynn et al., 2012*).

Interestingly, we observed large increases in the bound fraction of SHLD2 upon induction of DSBs, while there was no significant increase for SHLD1. This observation would be consistent with two potential models. First, SHLD1 may not bind every break where SHLD2 is present, but rather transiently visit breaks to facilitate Polα-dependent end fill-in via an interaction with CTC1 (*Mirman et al., 2022*). Another possible explanation is that SHLD1 only binds to one SHLD2 molecule at the 3' DNA end, while SHLD2 may accumulate either in excess of its substrate (potentially accumulating in 53BP1 condensates), or by forming SHLD2 homopolymers on the ssDNA overhang.

It is well-documented that SHLD3 is the furthest upstream factor in the Shieldin complex coordinating 53BP1-RIF1 with REV7-SHLD2-SHLD1. In experiments overexpressing GFP-SHLD2, knockdown of endogenous RIF1, 53BP1, or SHLD3 led to a marked decrease in GFP-SHLD2 recruitment (*Noordermeer et al., 2018*), which we confirmed by knocking out SHLD3 in the Halo-SHLD2 cell line. Our results demonstrate that similar quantities of SHLD3 and SHLD2 are simultaneously recruited to DNA lesions, consistent with the established recruitment hierarchy. In addition, SHLD3 recruitment continuous after SHLD2 recruitment is saturated, which further supports the model that the shieldin complex is not pre-assembled in the nucleoplasm.

## MDC1 and RIF1 are constitutive chromatin-binding proteins

Surprisingly, live-cell single-molecule imaging revealed constitutive chromatin-association of RIF1 and MDC1 in unperturbed cells. The fraction of chromatin bound MDC1 and RIF1 molecules is comparable to what we observe for Halo-H2B. While RIF1 and MDC1 are known to interact with chromatin, the absence of a substantial freely diffusing population leads to several questions for how these proteins function in DSB repair. First, if MDC1 and RIF1 are so immobile, how do they get recruited to DSBs? If the majority of MDC1 and RIF1 are always chromatin-associated, is the small pool of freely diffusing MDC1 and RIF1 sufficient for DSB repair or do chromatin bound molecules need to be locally reorganized or released from chromatin to form a focus? Additionally, RIF1 has many reported nuclear functions that require chromatin binding including telomere maintenance, 3D genome architecture, controlling replication timing, and DSB repair. It is unclear whether the static RIF1 that we observe contributes to one particular function or represents several distinct RIF1 activities. As such, our single-molecule imaging approach would be useful to test how disrupting RIF1 interactions by separation-of-function mutations alters its chromatin binding properties. For MDC1 we demonstrated that the constitutive chromatin association is mediated by its PST repeat domain which was recently reported to associate with the nucleosome acidic patch (*Salguero et al., 2019*). Some evidence suggests this domain is critical for promoting γH2AX-independent DSB repair, which would suggest that MDC1 constantly scans chromatin searching for DNA damage sites. While our observations would be consistent with that interpretation, it is also possible that MDC1 plays another important as yet unidentified role either in chromatin biology or specifically in DSB repair. Although rarely investigated, MDC1 is annotated as possessing 4 variants produced by alternative splicing, three of which we readily detect in roughly equal amounts. One of these variants has an exclusion of amino acids 1124–1410 constituting roughly 50% of the PST repeat region. Because endogenous genome editing preserves this normal splicing pattern, it is possible that this splice variant accounts for the freely diffusing MDC1 that we observe. Future studies will be required to address the role of the PST repeat region and the constitutive chromatin association of MDC1 in DNA repair.

## Using single-molecule live cell imaging to analyze chromatin binding of DNA repair factors

We have developed a new method to analyze the recruitment of DNA repair factors to chromatin and DNA lesions using live cell single-molecule imaging. DNA repair factors that display a significant

change in their chromatin bound fraction after DNA damage induction (e.g. DNA-PKcs, SHLD2, SHLD3, RNF169) can be studied using this approach. This approach is also amenable to performing residence time analysis of bound proteins which can be used to make meaningful insights into the affinity of DNA repair factors for their substrates in living cells. While we only tested zeocin induced DSBs in this study. It is possible that other means of inducing DNA lesions (e.g. PARP inhibitors, topoisomerase inhibitors) could result in significant changes in the chromatin bound fraction of other DNA repair factors. Importantly, due to our ability to detect individual molecules our approach is not limited to the analysis of DNA repair foci because it does not require the local accumulation of DNA repair factors to assess their chromatin binding. Previously, quantitative analysis of the recruitment of DNA repair factors to DNA lesions in living cells was limited to using laser microirradiation (LMI). LMI induces complex lesions, and the precise chemistry of the damaged DNA cannot be controlled. Our single-molecule imaging-based approach opens the door to quantitatively analyze the binding of DNA repair factors to DNA damage sites induced with a wide range of agents that lead to chemically well-defined DNA lesions. In addition, single-molecule imaging can directly measure kinetic parameters such as the dissociation rate of DNA repair factors from DNA damage sites, which will significantly advance our quantitative understanding of DNA repair in human cells.

In total, the results described in this study provide important insights into the protein abundance, diffusion dynamics, and recruitment kinetics to DNA breaks of a range of DNA repair factors. Our observations revealed that MDC1 and RIF1 constitutively associate with chromatin and demonstrate that the shieldin complex components are assembled in the context of DNA breaks but otherwise act independently. In addition, the collection of cell lines we have generated and the single-molecule imaging-based analysis of DNA repair factor recruitment to DNA breaks will be valuable tools for scientists studying DNA repair in human cells.

## Materials and methods

### Cell lines and cell culture

U2OS cells were obtained from the American Type Culture Collection (ATCC), were authenticated by STR profiling and tested negative for mycoplasma contamination. U2OS cells were cultured in RPMI containing 10% fetal bovine serum (FBS) and 100 units/mL penicillin and 100 μg/mL streptomycin in a humidified incubator maintained at 37 ° C with 5% $CO_2$. For live-cell imaging experiments, cells were plated onto 24-well glass bottom plates and imaging was conducted using $CO_2$-independent medium containing 10% FBS, 100 units/mL penicillin and 100 μg/mL streptomycin at 37 °C and 5% $CO_2$.

### Molecular cloning, plasmids, and genome editing

All gRNA, primer, and homology arm sequences used for cloning are included in *Supplementary file 1*. All gRNAs were cloned into BpiI-digested px330 backbone using standard procedures and ssDNA oligos were purchased from IDT. px330-U6-Chimeric_BB-CBh-hSpCas9 was a gift from Feng Zhang (Addgene plasmid #42230; http://n2t.net/addgene:42230; RRID:Addgene_42230). All homology-directed repair (HDR) donor plasmids were cloned using Gibson Assembly into pFastBac Dual backbone (Thermo Fisher, #10712024) linearized with HpaI. Gibson Assembly for each HDR donor consisted of three inserts including a left homology arm, right homology arm and an intervening sequence containing either the N-terminal or C-terminal HaloTag sequence. The N-terminal tag consists of a 3 x FLAG tag, an inverted SV40 promoter and Puromycin resistance cassette (Puro$^R$) flanked by LoxP sites (to select for edited cells), the HaloTag, and a TEV protease cleavage site, followed by a short peptide linker. Conversely, the C-Terminal tag consisted of a short peptide linker and TEV protease cleavage site, followed by the HaloTag and the 3 X FLAG epitope, with the Puro$^R$ cassette oriented 3′ to the 3 X FLAG tag. This approach allowed for selectable direct insertion of the HaloTag at the C-Terminus of a protein without the need for the additional step of Cre-Lox recombination to remove the Puro$^R$ cassette. Homology arms consisted of >250 bp homologous to the genomic DNA directly upstream and downstream of the Cas9 cleavage site and were either ordered as double-stranded gene fragments from IDT or PCR amplified from genomic DNA. MDC1 deletion mutants were generated using rationally designed PCR primers to amplify MDC1 cDNA. Cloning of the Halo-MDC1 deletion mutants was done by Gibson Assembly into pRK2 (Modified from pHTN HaloTag CMV-neo; Promega; #G7721). All plasmids were confirmed for proper insertion/assembly

by Sanger sequencing. Halo-H2B plasmid was kindly provided by Dr. Anders Hansen (*Hansen et al., 2017*) and the Halo-3xNLS plasmid was previously established and described (*Klump et al., 2023*).

Halo-MDC1 deletion mutants were expressed transiently by transfecting ~5 x $10^5$ cells with 1 μg of plasmid DNA using FuGene 6 (Promega). For genome editing, ~5 x $10^5$ U2OS cells were transfected using FuGene 6 with either 500 ng or 1 μg of each gRNA/Cas9 and HDR plasmid in 6 well plates. Approximately 2–3 days post-transfection, edited cells were selected for with puromycin (1 μg/mL). After puromycin selection, cells were allowed to grow for ~2–3 weeks followed by sorting for single-cell clones. N-terminally edited cells were transfected with a plasmid encoding Cre to recombine out the Puro$^R$ cassette generating a 3xFLAG-HaloTagged protein. Cells were labeled with JF646-HaloTag ligand (JF646) and sorted based on JF646 signal (*Grimm et al., 2015*). For knockout of Halo-MDC1, Halo-53BP1, and SHLD3, cells were transfected with two gRNA plasmids. For Halo-MDC1 and Halo-53BP1 cells were isolated for single-cell clones by labeling with JF646 and sorting JF646-negative cells. Knockout of Halo-MDC1 and Halo-53BP1 was confirmed by loss of fluorescence by SDS-PAGE, Western blot, Sanger sequencing and for MDC1 with Inference of CRISPR Edits (ICE), while knockout of SHLD3 was confirmed by genomic PCR (*Conant et al., 2022*). Knock-in of the DR-GFP HR reporter into the *AAVS1* locus was performed inΔMDC1 cells using AAVS1-DRGFP (Addgene plasmid #113193; RRID:Addgene_113193) and PX458-AAVS1 (Addgene plasmid #113194; RRID:Addgene_113194) which were kindly provided by Adam Karpf. After selection of edited cells with puromycin, cells were seeded by single-cell dilution in 96 well plates and clones were subsequently screened for DR-GFP insertion by genomic PCR.

## SDS-PAGE and western blot

Mini-PROTEAN TGX stain-free gels (BioRad) were used for SDS-PAGE for most proteins except for MDC1, DNA-PKcs, and RIF1 where homemade 6% polyacrylamide gels were used. Protein samples were made by lysing cells with 2 x Laemmli buffer with β-mercaptoethanol. For in-gel fluorescence to detect HaloTagged proteins, cells were labeled with ~150–500 nM JF646 or JFX650 HaloTag ligand (JFX650; similar to JF646 but with improved photostability) for 30 min, washed with complete medium three times, allowed to rest for ~10 min, and subsequently washed twice with PBS before being lysed with buffer (*Grimm et al., 2021*). For in-gel fluorescence measurements of protein abundance, cells were plated in triplicate in 24-well plates, the following day samples were labeled with 500 nM JF646 for ~30 min, washed with PBS three times, lysed in 2 x Laemmli buffer with β-mercaptoethanol, and boiled for 5 min at 95 °C prior to gel loading. Standards were prepared using standards containing known femtomolar concentrations of FPLC-purified, JF646-labeled 3X-FLAG-HaloTag prepared in aliquots containing known cell concentrations, boiled for five minutes, and frozen at –80 °C until use. Fluorescence was detected using the Cy5.5 filter on a BioRad Chemidoc. Stain-free detection of protein loading was detected using the Stain-Free filter on a BioRad Chemidoc after 45 second UV activation. For protein abundance, the number of molecules were calculated by comparing the fluorescent signal for each protein relative to the standard curves for cell number and JF646-labeled 3X-FLAG-HaloTag. For TEV corrections, ~120,000 cells were labeled with 500 nM JF646 for 30 min, washed once with PBS and harvested with 5 mM EDTA in PBS. Samples were lysed in 60 μL CHAPS lysis buffer and 20 μL incubated on ice with 5 units of TEV protease (New England Biolabs, P8112) for 30 min at which time 5 μL 6 x SDS sample buffer was added to 20 μL of the digested and undigested sample and boiled at 95 °C for 5 min. For TEV correction experiments where protease inhibitors used, samples were pre-treated with 10 μL of protease inhibitor cocktail (Sigma, P8340) for 30 min on ice followed by the addition of TEV protease. Comparisons between the JF646 signal ±digestion by TEV protease were performed by comparing the JF646 intensity for each sample after normalizing to stain-free protein signal. Transfers onto PVDF or Nitrocellulose membrane were conducted using either a Trans-Blot Turbo system (with Turbo transfer buffer) (BioRad) or by traditional wet tank transfer using CAPS Buffer with 10% Methanol (for DNA-PKcs, 53BP1, RIF1, and MDC1). Antibodies used were anti-FLAG-HRP (Sigma-Aldrich, A8592, RRID: AB_439702, 1:5000 dilution), anti-DNA-PKcs (a gift from Dr. Kathy Meek, 1:1000 dilution), anti-ATM (Santa Cruz, sc-135663, RRID: AB_2062962, 1:1000), anti-NBS1 (BioRad, VMA00403, 1:1000), anti-MDC1 (Novus Biologicals, NB100-395, RRID: AB_10001489, 1:1000), anti-RNF168 (GeneTex, GTX129617, RRID: AB_2886056, 1:1000), anti-53BP1 (Novus Biologicals, NB100-304, RRID: AB_10003037, 1:1000), anti-RIF1 (Bethyl Laboratories, A300-568A,

RRID: AB_669806, 1:1000), anti-REV7 (Abcam, ab180579, RRID: AB_2890174, 1:1000), goat anti-mouse HRP (Invitrogen, 31430, 1:2000), goat anti-rabbit HRP (Invitrogen, 31460, 1:2000).

## Purification of recombinant 3xFLAG-HaloTag protein

The plasmid encoding 6xHIS-3xFLAG-HaloTag protein (a kind gift from Dr. Daniel T. Youmans and Dr. Thomas R. Cech) was transformed into OneShot BL21(DE3) cells (Invitrogen). Cultures were grown to an OD600 of 0.6, induced with 1 mM IPTG, and grown at 18 °C for 16 hr shaking at 180 RPM. Bacterial cells were harvested by centrifugation at 5000x*g* for 15 min at 4 °C and frozen at –80 °C for 1 hr. Frozen cell pellets were resuspended in lysis and wash buffer (50 mM sodium phosphate buffer pH 8.0, 300 mM sodium chloride, 10 mM imidazole, 5 mM beta-mercaptoethanol). Cells were lysed by addition of 0.5 mg/ml lysozyme and sonication (40% amplitude, 90 s of sonication, 10 s pulses, 20 s pause, Fisherbrand Model 505, 0.5 inch tip) in an ice water bath. Cell lysates were cleared at 40,000x*g* and 4 °C for 30 min and incubated with fast flow nickel Sepharose (Cytiva) for 1 hr at 4 °C. The resin was washed three times with lysis and wash buffer and the 6xHIS-3xFLAG-HaloTag was eluted in elution buffer (50 mM sodium phosphate buffer pH 7.0, 300 mM sodium chloride, 250 mM imidazole, 5 mM beta-mercaptoethanol). The 6xHIS-3xFLAG-HaloTag was further purified using size exclusion chromatography using a superdex 75 column into 50 mM Tris pH 7.5, 150 mM potassium chloride, 1 mM DTT. Peak fractions were combined, concentrated, supplemented with 50% glycerol, snap frozen in liquid nitrogen and stored at –80 °C. To fluorescently label the 6xHIS-3xFLAG-HaloTag, we incubated the protein with a twofold excess of JF646 HaloTag-ligand overnight at room temperature. Excess fluorescent dye was removed by size exclusion chromatography using a superdex 75 column into 50 mM Tris pH 7.5, 150 mM potassium chloride, 1 mM DTT. The protein concentration and labeling efficiency was determined by absorption spectroscopy using $\varepsilon_{280nm} = 41,060$ M$^{-1}$ cm$^{-1}$ for the 6xHIS-3xFLAG-HaloTag (calculated using primary protein sequence and the ExPASy ProtParam tool) and $\varepsilon_{646nm} = 152,000$ M$^{-1}$ cm$^{-1}$ for the JF646 fluorescent dye (*Grimm et al., 2015*).

## Clonogenic survival assays

On the day prior to treatment cells were seeded in triplicate at a density of 500 cells per well in six-well plates. The following day, cells were treated with Zeocin (Gibco) at the indicated concentrations for four hours in complete medium. After treatment, medium was removed and replaced with fresh complete medium. For assays including the ATM inhibitor (ATMi) (KU-55933; Selleckchem), cells were pre-treated for 2 hr, during Zeocin treatment, and for 24 hr post-Zeocin with 10 µM ATMi. Cells were allowed to grow until colonies reached a size of >50 cells (approximately 7–10 days) at which time media was removed, wells washed with PBS and cells fixed and stained in crystal violet solution (20% ethanol and 1% w/v crystal violet). After incubation excess staining solution was removed and plates gently washed in diH2O and air-dried. Plates were imaged on a BioRad Chemidoc using the Coomassie Blue filter set. Colony counts were determined using ImageQuant TL 8.2. All colony survival assays were performed in triplicate at least three times and are plotted as the average ± S.D.

## Flow cytometry

Four independent experiments were performed to quantify relative protein abundance using flow cytometry. On the day prior to performing flow cytometry, ~100,000 cells of each HaloTagged protein clone were seeded into 24-well plates. The next day, cells were labeled with 500 nM JF646 for ~30 min, washed three times with PBS, and allowed to rest for 5 min in complete medium to allow unbound dye to leak out of the cells. Cells were harvested using 5 mM EDTA in PBS, fixed in 2% paraformaldehyde for 10 min, washed once with PBS and resuspended in PBS with 1% bovine serum albumin. Samples were run on a Cytek Aurora spectral flow cytometer and >7500 events were collected per protein for each replicate to determine mean fluorescence intensity. Data were analyzed in FCS Express 7. For DR-GFP experiments, ~200,000 cells were nucleofected in RPMI plus 50 mM HEPES with 400 ng I-SceI plasmid and 600 ng of each HaloTagged MDC1 or 3XFLAG-Halo-NLS plasmid and seeded in six-well plates. Forty-eight hr post-nucleofection, cells were labeled with 150 nM JFX650, collected by trypsinization, and samples were run on a BD Accuri C6 cytometer and >5000 JFX650-positive cells were collected for each sample to determine the percentage of GFP-positive cells. Data were analyzed in FCS Express 7.

## Laser microirradiation

Laser microirradiation (LMI) was carried out on a Olympus IX83 inverted microscope equipped with a 4-line cellTIRF illuminator (405 nm, 488 nm, 561 nm, 640 nm lasers), an Excelitas X-Cite TURBO LED light source, a Olympus UAPO 100 x TIRF objective (1.49 NA), a CAIRN TwinCam beamsplitter, 2 Andor iXon 897 Ultra EMCCD cameras, a cellFRAP with a 100 mW 405 nm laser, an environmental control enclosure and operated using the Olympus cellSense software. Cells were seeded the day before LMI onto a 24-well glass bottom plate. On the day of the experiment, cells were labeled with 150 nM JFX650 for 30 min, washed three times with complete media and presensitized with Hoechst (1 µg/mL) for 10 min before washing and adding fresh complete medium. Plates were placed on the microscope stage which was pre-warmed to 37 °C with 5% $CO_2$. Cells were irradiated using a 20ms pulse at 25% laser power using drawn interpolated lines or a diffraction limited spot (only used for REV7 due to diffuse staining pattern which made it difficult to quantify over time). Images were acquired using a 100 x objective and fluorescence imaged with excitement by the 630 nm LED light source at various rates (e.g. every 1 s, 2 s, 5 s, or 10 s). To quantify the fluorescent signal after LMI, we converted movies to.tif files. Irradiated cells were cropped and images drift corrected using NanoJ in Fiji (*Laine et al., 2019*). Using drift corrected movies, an ROI was placed around the irradiated area and we measured the mean intensity within the ROI over time. Baseline fluorescence was subtracted from each sample in order to calculate the relative increase in fluorescence that accumulated over time. To normalize fluorescence intensity values, each cell had its highest intensity frame set to one in order to normalize fluorescence intensity values between samples. For averaging normalized intensity values for all cells, the frame number with the highest average intensity was set to one. Data were plotted in GraphPad Prism and recruitment half-time was determined by fitting with an exponential one-phase association model ($Y=Y0 + (Plateau − Y0)*(1-exp(-K*x))$) where Y0 was set to zero and Plateau was set to one.

## Live-cell imaging

Live-cell and live-cell single-molecule imaging was performed on the same microscope described above. For live-cell imaging of HaloTagged DDR protein localization and DNA-damage induced foci, samples were densely labeled with 500 nm JF646 for 15–30 min, washed three times with complete medium and allowed to rest for 10 min before adding $CO_2$-independent medium. For Zeocin-treated samples, cells were treated for one hour prior to HaloTag labeling with 100 µg/mL Zeocin. Z-stack images were acquired at 37 °C in the presence of 5% $CO_2$ with a 100 x objective and the 640 nm laser. These experiments were performed twice imaging >20 cells per experiment. For live-cell single-molecule imaging, HaloTagged DDR proteins were labeled with JFX650 at differing concentrations to achieve single-molecule density (0.1 nM for 30 s up to 20 nM for 1 min). After labeling, cells were washed three times with complete medium and allowed to rest for 10 min before adding $CO_2$-independent medium. For Zeocin-treated samples, cells were treated for one hour with Zeocin (100 µg/mL) prior to protein labeling with JFX650. Imaging was performed at 37 °C and 5% $CO_2$ with the 100 x objective and the 640 nm laser with highly inclined laminated optical sheet (HILO) illumination (light angled between 1.29 and 1.32 depending on the sample). Images were acquired at 138 fps for 3000 frames followed by a brightfield image to visualize the cell. For dual color imaging of Halo-53BP1, cells were first labeled with 2.5 nM JFX650 for one minute followed by 100 nM JF503 for 10 minutes. Images were acquired at 202 fps and collected using a 60 x objective and the 640 nm and 488 nm laser lines. Imaging of MDC1 mutants in the presence of absence of Zeocin were acquired at 202 fps with a 60 x objective and the 640 nm laser line with HILO illumination.

## Analysis of live-cell single-molecule imaging

Single-particle tracking (SPT) was performed in MATLAB 2019a using a version of SLIMfast allowing for analysis of TIFF files (*Hansen et al., 2018*). Settings for SPT used in the analysis were as follows: Exposure Time = 7.24ms (4.95ms for data presented in *Figure 7—figure supplement 1C*), NA = 1.49, Pixel Size = 0.16 µm (0.1083 for data presented in *Figure 7—figure supplement 1C*), Emission Wavelength = 664 nm, $D_{max}$ = 5 µm²/s, Number of gaps allowed = 2, Localization Error = –5, Deflation Loops = 0. While most HaloTagged DDR proteins exhibited near complete nuclear localization, REV7, SHLD3, SHLD2, and SHLD1 possessed mixed cytoplasmic and nuclear localizing fractions. For these proteins, the brightfield image was used to generate a nuclear mask in FIJI to separate cytoplasmic

from nuclear tracks. Only nuclear tracks were used for analysis of protein diffusion in SpotOn. SPT files were then used in SpotOn in MATLAB to determine diffusion coefficients and the percentage of bound versus free particles. The following settings were used in SpotOn analysis of SPT files: TimeGap = 7.24ms (4.95ms for data presented in *Figure 7—figure supplement 1C*), dZ = 0.700 µm, GapsAllowed = 2, TimePoints = 8, JumpsToConsider = 4, BinWidth = 0.01 µm, PDF-fitting, D_Free_2State = [0.5 25], D_Bound_2State = [0.0001 0.5]. All live-cell single molecule imaging experiments were performed three times acquiring >20 cells per condition per experiment. Comparisons of Diffusion coefficients and fractions bound were performed in GraphPad Prism by two-way ANOVA with Tukey's posthoc test. To filter out trajectories that overlapped with DNA repair foci, we generated a mask using the quantitatively labeled Halo-53BP1 signal and assigned Halo-53BP1 (JFX650) trajectories whose coordinates overlapped with the mask for at least on frame to the DNA repair foci group.

### Residence time analysis

For the residence analysis of low mobility particles, we averaged the intensity of each pixel of single-molecule live cell imaging movies over 10 consecutive frames reducing the effective imaging rate from 138 to 13.8 frames per second. This approach blurs out the signal from mobile molecule because they do not reside in the same location for multiple consecutive frames. Single particle tracking was then carried out constraining $D_{max}$ to 0.5 µm²/s to exclusively track low mobility molecules, which are assumed to be chromatin bound. Track length distributions were plotted as survival probabilities (1 – Cumulative density function of the track lengths). To determine the rate constants and corresponding half-lives the decay functions were fit two a two-component exponential decay function.

### Immunofluorescence

Two days prior to performing immunofluorescence, ~1 million ΔMDC1 cells were nucleofected with two mg of Halo-MDC1 WT, ΔPST, or ΔBRCT plasmid using a Lonza 4D nucleofector. The following day, ~250,000 cells were seeded onto glass coverslips in six-well plates in complete medium. The next day, cells were left untreated or treated with 10 mg/mL Zeocin for one hour after which samples were labeled with 150 nM JF646 HaloTag ligand for 10 min, washed three times, and incubated in complete medium for 10 min to allow unbound ligand to leak out of the cells. Next, cells were washed with PBS and fixed in 4% formaldehyde prepared in PBS for 10 min at room temperature. After fixation, coverslips were washed with PBS and cells permeabilized with 0.2% Triton-X 100 for 2 min. Next, coverslips were washed twice with antibody dilution buffer (ABDIL) (3% BSA in PBS-Tween 20 (PBS-T)) and incubated in ABDIL for one hour. Coverslips were incubated with anti-53BP1 primary antibody (Cell Signaling Technology Cat# 4937, RRID:AB_10694558) in ABDIL at 1:200 dilution for 1 hr, followed by 35-min washes with PBS-T, and 1 hr incubation with goat anti-rabbit antibody conjugated to AF488 (Thermo Fisher Scientific Cat# A-11034, RRID:AB_2576217) at 1:500 dilution for 1 hr. After washing three times with PBS-T, DNA was stained with Hoechst, coverslips mounted onto slides with ProLong Diamond Antifade Mountant and sealed with nail polish. Samples were imaged using a DeltaVision Elite 642 microscope with a 60 x PlanApo objective (1.42 NA) and a pco.edge sCMOS camera. Images were processed by deconvolution in DeltaVision Softworx followed by maximum intensity projection in ImageJ.

### Acknowledgements

We are grateful to Dr. Kathy Meek for providing the DNA-PKcs antibody. We thank Dr. Daniel T Youmans and Dr. Thomas R Cech for providing the plasmid for recombinant production of the 3xFLAG-HaloTag protein. We thank Dr. Eric Patrick for contributing to the purification of the 3xFLAG-HaloTag protein. Funding This work was funded, by NIH grants F32GM139292 to JRH and DP2GM142307 to JCS. This work was supported by the microscopy and flow cytometry cores in the Institute of Quantitative Health Sciences and Engineering at Michigan State University. The MSU Flow Cytometry Core facility is funded, in part, through the financial support of Michigan State University's Office of Research & Innovation, College of Osteopathic Medicine, and College of Human Medicine.

## Additional information

### Funding

| Funder | Grant reference number | Author |
| --- | --- | --- |
| National Institutes of Health | F32GM139292 | Joshua R Heyza |
| National Institutes of Health | DP2GM142307 | Jens C Schmidt |

The funders had no role in study design, data collection and interpretation, or the decision to submit the work for publication.

### Author contributions

Joshua R Heyza, Conceptualization, Formal analysis, Funding acquisition, Investigation, Methodology, Writing - original draft, Writing – review and editing; Mariia Mikhova, Formal analysis, Investigation, Writing – review and editing; Aastha Bahl, Investigation; David G Broadbent, Formal analysis, Investigation, Methodology; Jens C Schmidt, Conceptualization, Formal analysis, Funding acquisition, Methodology, Project administration, Writing – review and editing

### Author ORCIDs

Joshua R Heyza  http://orcid.org/0000-0003-0847-0501
David G Broadbent  http://orcid.org/0000-0002-0940-1068
Jens C Schmidt  http://orcid.org/0000-0001-9061-7853

### Decision letter and Author response

Decision letter https://doi.org/10.7554/eLife.87086.sa1
Author response https://doi.org/10.7554/eLife.87086.sa2

## Additional files

### Supplementary files

• Supplementary file 1. Oligonucleotide sequences used in this study including gRNAs, PCR primers, homology arms, and N-terminal and C-terminal 3XFLAG-HaloTag.

• MDAR checklist

### Data availability

All uncropped images for gels and blots included in this manuscript have been provided as source data.

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
