## [Editor Report]

This manuscript reports valuable tools and data to study DNA repair and its regulation in life cells by generating and validating cell lines with Halo-tag fusions to the chromosomal genes encoding ATM, NBS1, MDC1, RNF168, RNF169, 53BP1, RIF1, SHLD3, REV7, SHLD2, SHLD1, and DNA-PKcs. The data establish the utility of most of the tools but remain incomplete. Conclusions from the kinetic analysis would benefit from more validation by genetic experiments and the single particle tracking analysis offers more potential for analysis.

---

## [Decision Letter]

**Decision letter after peer review:**

[Editors’ note: the authors submitted for reconsideration following the decision after peer review. What follows is the decision letter after the first round of review.]

Thank you for submitting the paper "Systematic analysis of the molecular and biophysical properties of key DNA damage response factors" for consideration by *eLife*. Your article has been reviewed by 3 peer reviewers, including Wolf-Dietrich Heyer as the Reviewing Editor and Reviewer #1, and the evaluation has been overseen by a Senior Editor. The following individuals involved in the review of your submission have agreed to reveal their identity: Markus Löbrich (Reviewer #2); Judith Miné-Hattab (Reviewer #3).

Comments to the Authors:

We are sorry to say that, after consultation with the reviewers, we have decided that this work will not be considered further for publication by *eLife*.

The reviewers and Reviewing Editor recognize the potential utility of the reported tools for the scientific community and appreciate that these tools can provide certain novel insights. However, the analysis is incomplete in several areas and the conclusions seem insufficiently supported by the experimental evidence. The potential revisions would be extensive and consume more time than compatible with *eLife*'s editorial policy.

If you decide to address the extensive revisions requested, we would encourage resubmission as a new manuscript, and we would make an effort to recruit the same reviewers to assess the work, which would be treated as a new submission. The major issues are the functionality of the tagged proteins, especially MDC1 and SHLD1/2 for which major conclusions are reached (#3), completeness of the single particle tracking analysis (#5), and orthogonal genetic validation of the major conclusions from the kinetic analysis (#11, 13, 16).

Life cell imaging provides unprecedented insights into cellular processes, and advances in fluorescence and microscopy allow the identification and tracking of single protein particles in four dimensions. The manuscript reports the creation of a set of useful cell lines with Halotag fusions to 12 key proteins acting in the DNA damage response, namely ATM, NBS1, MDC1, RNF168, RNF169, 53BP1, RIF1, REV7, SHLD1/2/3, and DNA-PKcs. The fusions were carefully validated molecularly and functionally, leading to detectable expression of Halotagged proteins in protein gels. All proteins, with the exception of ATM-Halo, showed the expected cellular localization in undamaged cells and led to an increase in focus formation in response to DNA damage (Zeocin) with the exceptions of ATM and DNA PKcs. Clonogenic survival assays demonstrated the functionality of the fusion proteins, with the exception of Halo-ATM, which appeared similar to a loss of function for both a C- and an N-terminal fusion, and Halo-MDC1, Halo-SHLD1/2, and Halo-53BP1, which showed partial loss of function. Using flow cytometric and in-gel approaches, the steady protein level for all Halo fusions was determined with a good correlation between both methods. All experiments were conducted with two independent clones for each fusion (except NBS, SLD2, 53BP1). All cell lines were homozygous for the Halo tag, with the exception of Halo-SHLD2, where one allele was tagged and the other was a frameshift allele.

Kinetic recruitment experiments using the Halo-tagged proteins were conducted using laser microirradiation-induced DNA damage. The data for components of the Shieldin complex showed significantly different recruitment kinetics for SHLD2 and 3, suggesting that this protein complex assembles at the site of DNA damage rather than being pre-assembled. The kinetic resolution for the other proteins allowed the identification of additional differences in protein recruitment at laser-induced DNA damage. The caveat working with non-physiological laser-induced DNA damage could have been considered and potentially be selectively complemented with orthogonal ways to induce DNA damage such as Cas9-mediated DSBs.

Single particle tracking was used to determine the nuclear diffusion of the single proteins in the presence and absence of DNA damage (zeocin). It is unclear if the technology allows the tracking of a single molecule. The results identified significant differences between the Shieldin subunits 1, 2, and 3, corroborating the conclusion that they do not exist as a pre-assembled complex. The data for MDC1 and RIF1 suggest that both are largely chromatin-associated in undamaged cells, and follow-up experiments show that the MDC1 PST domain is responsible for this, whereas the BRCT domain of MDC1 is critical for DNA damage-induced chromatin association, as previously shown. The analysis of the MDC1 domains lacks experiments under DNA damage conditions (zeocin).

Overall, this manuscript is well-written and documented but the analysis remains incomplete in several instances.

Recommendations for the authors:

1) The ATRX mutation in U2OS cells affects DNA repair pathway choice (PMID: 29937341, PMID: 33431668). This caveat should be considered and discussed.

2) I do not understand the comment in lines 210/211: "Importantly, the differences in absolute protein number between independent genome-edited clones could be the consequence of a different number of alleles being modified with the HaloTag." It is stated in lines 115-117 that all lines are homozygously tagged except for one, where the other allele is a frameshift likely resulting in expressing an unstable truncation protein. There should be no variation in tagged alleles, or am I missing something?

3) The survival of cells expressing Halo-tagged version of ATM, MDC1, 53BP1, SHLD1, and SHLD2 is reduced after zeocin treatment compared to wild-type cells. Thus, the functionality of these tagged proteins is questionable. In particular, the analysis focuses on MDC1 although MDC1 is one of the less functional tagged proteins. For example, MDC1 is reported as less mobile than histones H2B: is this really possible? Could it be an effect of the partial loss of MDC1 functionality? If not, how can such slow mobility be explained? The manuscript also reports that the constitutive interaction between MDC1 and chromatin is mediated by the PST repeat domain of MDC1: again, it is necessary to be careful about this conclusion since the functionality is reduced in the Halo-tagged version.

We appreciate that fully functional fusions may be out of reach, but the limitations need to be acknowledged and discussed. Have alternative tag designs been tried?

4) In the section: Functional validation of HaloTagged DDR proteins:

In the absence of zeocin treatment, cells expressing Halo-MDC1 exhibit many spontaneous foci. Is it something known and if not, how can that be explained?

Cells expressing ATM-Halo do not form foci after zeocin treatment: please comment.

Concerning DNA-PK: this protein is highly abundant in the cell; however, it does not form foci after zeocin treatment. Is there an explanation? does it mean that even if the protein is very abundant, very few DNA-PK molecules are present within foci and sufficient for the next steps of NHEJ? Does it form a visible line after micro-irradiation?

5) Page 12, line 206: It is stated that MDC1 has a higher protein abundance than ATM, SHLD1, SHLD2, and SHLD3 but one of the two MDC1 clones analyzed has the lowest protein abundance of all clones analyzed in this study (2400 molecules per cell according to the text and table I). Why are the two MDC1 clones differ so drastically from each other? Confusingly, the big difference between the two clones is not seen in the bar chart in Figure 3B and is not measured by flow cytometry.

6) Page 12, line 231: It is stated that the adjustment factors for DNA-PKcs are 0.62 and 0.79 but the application of these factors increases the molecule number for clone 1 but decreases it for clone 2 (the latter seems wrong).

7) Single Particle Tracking analysis:

Using analysis of single particle tracking, the authors can measure the diffusive properties of repair proteins. Diffusion is estimated in the presence and in absence of zeocin treatment. Thus, the cells contain many foci: all the traces in the nucleus are analyzed at once, inside and outside foci. The authors then used a 2 population model to fit the distribution of protein displacements.

The SPT analysis allows the authors to give some interesting mechanistic insight but the authors could extract much more information from their data. Why are the values of Dslow not provided? The authors interpret the slow population as bound to damaged DNA. However, it is known that some proteins diffuse relatively fast inside repair foci, especially if they are able to form liquid-liquid phase separation. In addition, some proteins might diffuse slowly outside of foci, because of non-specific interactions (or even for MDC1 for example). Thus, it is possible that the slow molecules are a mixture of the molecules inside the foci with molecules exhibiting chromatin binding outside of the repair foci.

There is no visualization of the traces allowing us to see if the slow molecules are indeed inside foci and the fast ones are outside. It would be essential to be able to see this for at least 1 cell for each repair protein.

Are mixed traces observed, with a slow and a fast part?

Is it possible to estimate the residence time of each protein inside repair foci or on their substrate?

Are the proteins inside foci exchanging with the rest of the nucleus or are they stuck inside the focus during the entire trace?

Since you use bright JF, it should be possible to have long traces: the authors should show a distribution of the traces' length for each repair protein.

Do you see a change in protein diffusion in the absence of zeocin treatment and in the presence of zeocin treatment outside of foci?

The authors also use H2B and NLS as controls. The values obtained should be compared with values found in the literature.

Finally, it is not clear how the histones H2B are tagged in this study: is it also an endogenous H2B-Halo tag? Is it a stable cell line but not endogenous, or is it a transient transfection?

8) Laser micro-irradiation induces massive damage and may not be reflective of physiological encountered DNA damage. Have the authored considered using Cas9-induced DSBs as a defined and targeted DNA damage? I understand that adding such experiments for all proteins would be a massive endeavor, but maybe this could be done for MDC1 and/or the Shieldin complex. Regardless, the limitations of the laser micro-irradiation approach should be discussed.

9) Figure 6 lacks data for the analysis in the presence of zeocin, in the way it was done in the analysis for Figure 5. Such data will corroborate the foci analysis and potentially reveal differences in the recruitment of MDC1 to damaged sites.

10) What is the evidence that a single molecule can be tracked in Figures 5 and 6, as opposed to a single particle that may be composed of multiple proteins?

11) Page 13, line 256: It was surprisingly observed that REV7 and SHLD3 have vastly different recruitment times to laser-induced damage although both factors are known to interact. At the end of the Discussion section, it is speculated that this might reflect REV7's role in TLS but the reader would benefit if such an interpretation was offered earlier in the paper. Moreover, if this interpretation was correct, one would expect that REV7 was recruited to laser damage independently of SHLD3. This should be tested by siRNA-mediated depletion of SHLD3 (or other factors upstream of SHLD3).

12) Page 14, lines 260-263: It is observed that SHLD1 is not recruited to laser damage although it forms foci after zeocin treatment. The authors then speculate that this might suggest that SHLD1 recruitment is the key regulatory step for a fill-in reaction. This is not clear. Are the authors suggesting that fill-in takes place at zeocin-induced breaks but not at laser-induced breaks? Please clarify.

13) Page 14, line 266: The authors state that the measured recruitment kinetics provide insight into the interdependencies of the shieldin complex components. As evidenced by the very early recruitment kinetics of REV7 (possibly due to its role in TLS), this may not be true since the factors could have roles in other processes. Thus, the evaluation of interdependencies requires the measurement of the recruitment kinetics in situations when individual components of the shieldin complex are depleted by siRNA technology. Thus, the authors should assess the recruitment kinetics in cells depleted for SHLD2/3, REV7, or 53BP1/RIF1.

14) Page 17, line 319: SHLD1 is not recruited to chromatin after DSB induction despite its low abundance, and the authors state that this is consistent with its lack of recruitment to laser damage. However, it is clearly shown to form foci at DSB. Moreover, how does this finding fit the author's suggestion from above that SHLD1 is the key regulator of the fill-in reaction?

15) The authors try to address the unexpected findings for SHLD1 (no recruitment to laser damage and no increase in the chromatin-bound fraction after zeocin treatment) in the discussion on page 22, last paragraph, and suggest that SHLD1 may either not bind every DSB where SHLD2 is present or may reside at a DSB in lower copy numbers than SHLD2. Both explanations appear inconsistent with the robust formation of SHLD1 foci at zeocin-induced DSB, the first suggestion can easily be tested with co-localization experiments using HaloTag SHLD1 and SHLD2 ab or vice versa.

16) Page 21, line 390: The authors suggest the model that RNF169 delays 53BP1 recruitment to DSBs. Although this interpretation is consistent with the presented data, the model should be tested by 53BP1 recruitment measurements after siRNA-mediated depletion of RNF169.

[Editors’ note: further revisions were suggested prior to acceptance, as described below.]

Thank you for resubmitting your work entitled "Systematic analysis of the molecular and biophysical properties of key DNA damage response factors" for further consideration by *eLife*. Your revised article has been evaluated by Detlef Weigel (Senior Editor) and a Reviewing Editor.

The manuscript has been improved but there are some remaining issues that need to be addressed, as outlined below:

Life cell imaging provides unprecedented insights into cellular processes, and advances in fluorescence and microscopy allow the identification and tracking of single protein particles in four dimensions. The manuscript reports the creation of a set of useful cell lines with Halotag fusions to 12 key genes acting in the DNA damage response, namely ATM, NBS1, MDC1, RNF168, RNF169, 53BP1, RIF1, REV7, SHLD1/2/3, and DNA-PKcs. The fusions were carefully validated molecularly and functionally, leading to detectable expression of Halotagged proteins in protein gels. All proteins, with the exception of ATM-Halo, showed the expected cellular localization in undamaged cells and led to an increase in focus formation in response to DNA damage (Zeocin) with the exceptions of ATM and DNA PKcs. Clonogenic survival assays demonstrated the functionality of the fusion proteins, with the exception of Halo-ATM, which appeared similar to a loss of function for both a C- and an N-terminal fusion, and Halo-MDC1, Halo-SHLD1/2, and Halo-53BP1, which showed partial loss of function. Using flowcytometric and in gel approaches, the steady protein level for all Halo fusions was determined with good correlation between both methods. All experiments were conducted with two independent clones for each fusion (except NBS, SLD2, 53BP1). All cell lines were homozygous for the Halo tag, with the exception of Halo-SHLD2, where one allele was tagged and the other was a frameshift allele.

Kinetic recruitment experiments using the Halo-tagged proteins were conducting using laser microirradiation induced DNA damage. The data for components of the Shieldin complex showed significantly different recruitment kinetics for SHLD2 and 3, suggesting that this protein complex assembles at the site of DNA damage rather than being pre-assembled. The kinetic resolution for the other proteins allowed identification of additional differences in protein recruitment at laser-induced DNA damage. The caveat working with non-physiological laser-induced DNA damage could have been considered and potentially be selectively complemented with orthogonal ways to induce DNA damage such as Cas9-mediated DSBs.

Single particle tracking was used to determine the nuclear diffusion of the single proteins in the presence and absence of DNA damage (zeocine). It is unclear if the technology allows tracking of a single molecule. The results identified significant differences between the Shieldin subunits 1, 2, and 3, corroborating the conclusion that they do not exist as a preassembled complex. The data for MDC1 and RIF1 suggest that both are largely chromatin-associated in undamaged cells, and follow-up experiments show that the MDC1 PST domain is responsible for this, whereas the BRCT domain of MDC1 is critical for DNA damage-induced chromatin association, as previously shown. The analysis of the MDC1 domains lacks experiments under DNA damage conditions (zeocine).

The two main conclusions from the work are that (1) the individual subunits of the Shieldin complex are recruited independently to sites of DNA damage, and (2) MDC1 and RIF1 are bound constitutively to chromatin. Although these conclusions are supported by the data, some inconsistencies with the literature remain unresolved. For example, the authors report that SHLD2 is recruited to laser tracks before SHLD3, while previous work demonstrated a genetic requirement for SHLD3 to recruit SHLD2 to sites of DNA damage. Further, it remains an open question how and when MDC1 and RIF1 are recruited to sites of DNA damage if they are constitutively bound to chromatin.

The use of Halo ligands with different emission spectra to simultaneously monitor single-particles and DNA repair foci is very elegant and can potentially be used to distinguish the behavior of different subpopulations of a DNA repair factor.

The manuscript is well-written, but some literature reference and information in the methods section are missing.

In conclusion, this is an interesting study that applies novel microscopy techniques to DNA double-strand break repair proteins. However, the study remains somewhat descriptive, which limits the mechanistic insight gained from the study and the analysis remains incomplete in several instances for lack of genetic corroboration of the main conclusion about the Shieldin complex recruitment.

Recommendations for the authors

Essential revisions

1) Line 144: The authors conclude that "the HaloTag does not impact the proper cellular localization of these proteins" based on fluorescence microscopy of the Halo-tagged proteins after JF646 labeling. This conclusion cannot be made, because it would require examination of the localization of the untagged proteins under the same conditions. Please qualify your statement.

2) Line 146: The authors conclude that "HaloTagging ATM at the N-terminus led to nuclear exclusion". The authors cannot make this conclusion without imaging the untagged ATM under the same condition. Please qualify your statement.

3) Line 183: It cannot be concluded that "most possess full DNA repair functionality" unless the cell lines expressing Halo-tagged proteins are compared to their gene knockout counterparts. This was only performed for 53BP1 and MDC1, where partial functionality was observed. Please qualify your statement.

4) Page 13, line 256: It was surprisingly observed that REV7 and SHLD3 have vastly different recruitment times to laser-induced damage although both factors are known to interact. At the end of the Discussion section, it is speculated that this might reflect REV7's role in TLS but the reader would benefit if such an interpretation was offered earlier in the paper. Moreover, if this interpretation was correct, one would expect that REV7 was recruited to laser damage independently of SHLD3. This should have been tested by siRNA-mediated depletion of SHLD3 (or other factors upstream of SHLD3) and this limitation should be explicitly acknowledged in the text.

5) Page 14, line 266: The authors state that the measured recruitment kinetics provide insight into the interdependencies of the shieldin complex components. As evidenced by the very early recruitment kinetics of REV7 (possibly due to its role in TLS), this may not be true since the factors could have roles in other processes. Thus, the evaluation of interdependencies requires the measurement of the recruitment kinetics in situation when individual components of the shieldin complex are depleted by siRNA technology. Thus, the authors should assess the recruitment kinetics in cells depleted for SHLD2/3, REV7 or 53BP1/RIF1. The caveat of roles in independent processes should be explicitly mentioned.

6) Line 293: It is counter-intuitive that SHLD2 foci depends on SHLD3, which is recruited to foci significantly later than SHLD2. The two-step model suggested in the Discussion to explain this observation is not supported well by the data, because one would expect the initial (pre-SHLD2) step of SHLD3 to also be detected in the LMI experiments. Further, SHLD2 actually dissociates when SHLD3 associates with laser stripes, which is not discussed by the authors. It would strengthen the manuscript if the model could be tested experimentally by the authors.

7) A similar study was conducted previously (Aleksandrov et al. 2018), which reported recruitment half-times to laser stripes significantly different than the current study for some proteins and in other cases similar half-times. For example, the recruitment times for MDC1, RNF168, RNF169, and 53BP1 was reported by Aleksandrov to be 35s, 78s, 203s, and 307s, where the current study report 77s, 69s, 186s, 669s. It would be in place to reference the previous study and compare findings.

8) It is surprising that Halo-H2B only displays 66% chromatin binding. The assumption is that JF646 binds irreversibly to the Halo-tag, but is it possible that some free JF646 is present and gives rise to the "free" pool of fluorophores? This could easily be tested by formaldehyde fixation which should give 100% chromatin binding if no free JF646 is present.

9) Line 331: A two-state diffusion model is assumed where particles either freely diffuse or are chromatin bound. How would the conclusions be affected if a third state was allowed where a protein is part of a slow diffusing macromolecular complex.

10) Figure 5: The scatter plots should be displayed as "super plots" where data points for each of the 3-4 independent experiments are presented in different colors/symbols (Lord et al. 2020). This would reveal any systematic differences between experiments. For example, it looks like RNF169 data points can be divided into two populations.

11) Page 21, line 390: The authors suggest the model that RNF169 delays 53BP1 recruitment to DSBs. Although this interpretation is consistent with the presented data, the model could be tested by 53BP1 recruitment measurements after siRNA-mediated depletion of RNF169. This limitation should be explicitly acknowledged in the text.

12) The sources of several plasmids are missing e.g. pX330 in line 665.

13) A table with oligonucleotides and gRNAs used in the study should be included.

14) Line 511: Which data allow the authors to conclude that the PST domain of MDC1 "facilitates DDR signal amplification"?

15) The authors conclude that MDC1 and RIF1 are constitutively associated with chromatin. If this is the case then one might expect the localization of MDC1 and RIF1 to follow that of condensed chromosome when cells progress from interphase into mitosis. Indeed, there is evidence for this for RIF1 (Watts et al. 2020), but for MDC1 I could not find such evidence in the literature. However, I suspect that the authors in their data have images of mitotic cells that could answer this question.

16) To quantify the mobility inside foci versus outside foci, and the flux of proteins in and out of foci, would it be possible to make a density map of all the repair proteins in the nucleus? Using this density map, the foci appear clearly, and it is then possible to separate trajectories within foci from those outside foci.

17) Figure S5B: After zeocin treatment: is it possible that the lines between foci are misslinking?

References

Aleksandrov R, Dotchev A, Poser I, Krastev D, Georgiev G, Panova G, Babukov Y, Danovski G, Dyankova T, Hubatsch L et al. 2018. Protein Dynamics in Complex DNA Lesions. Mol Cell 69: 1046-1061 e1045.

Lord SJ, Velle KB, Mullins RD, Fritz-Laylin LK. 2020. SuperPlots: Communicating reproducibility and variability in cell biology. J Cell Biol 219.

Watts LP, Natsume T, Saito Y, Garzon J, Dong Q, Boteva L, Gilbert N, Kanemaki MT, Hiraga SI, Donaldson AD. 2020. The RIF1-long splice variant promotes G1 phase 53BP1 nuclear bodies to protect against replication stress. *eLife* 9.

---

## [Author Response]

[Editors’ note: the authors resubmitted a revised version of the paper for consideration. What follows is the authors’ response to the first round of review.]

Essential revisions:1) The ATRX mutation in U2OS cells affects DNA repair pathway choice (PMID: 29937341, PMID: 33431668). This caveat should be considered and discussed.

We thank the reviewers for the comment regarding ATRX. The two references supplied do demonstrate that ATRX is not expressed in U2OS cells, however the Cancer Cell Line Encyclopedia characterization of human cancer cell lines using next-generation sequencing did not detect an ATRX mutation in U2OS cells, so the absence of expression may be due to another underlying cause (PMID: 31068700). We have now included a statement regarding the absence of ATRX expression in U2OS cells which plays important roles in DSB repair.

2) I do not understand the comment in lines 210/211: "Importantly, the differences in absolute protein number between independent genome-edited clones could be the consequence of a different number of alleles being modified with the HaloTag." It is stated in lines 115-117 that all lines are homozygously tagged except for one, where the other allele is a frameshift likely resulting in expressing an unstable truncation protein. There should be no variation in tagged alleles, or am I missing something?

While genomic PCR in Figure 1 demonstrates that all alleles are homozygously edited, these data do not report on how many alleles are present within each clone. Because cancer cell lines can possess inherent variations in karyotype within a cell population, we can not exclude the possibility that differences in relative expression of tagged proteins could be either a consequence of either a different number of alleles present in the clone that was genetically modified or a result of inherent clonal variations in expressing the protein of interest. We have now included a statement clarifying these two possibilities in the text.

3) The survival of cells expressing Halo-tagged version of ATM, MDC1, 53BP1, SHLD1, and SHLD2 is reduced after zeocin treatment compared to wild-type cells. Thus, the functionality of these tagged proteins is questionable. In particular, the analysis focuses on MDC1 although MDC1 is one of the less functional tagged proteins. For example, MDC1 is reported as less mobile than histones H2B: is this really possible? Could it be an effect of the partial loss of MDC1 functionality? If not, how can such slow mobility be explained? The manuscript also reports that the constitutive interaction between MDC1 and chromatin is mediated by the PST repeat domain of MDC1: again, it is necessary to be careful about this conclusion since the functionality is reduced in the Halo-tagged version.We appreciate that fully functional fusions may be out of reach, but the limitations need to be acknowledged and discussed. Have alternative tag designs been tried?

We have now included statistical analysis comparing the survival curves for each protein depicted in Figure 2A. The reviewers are correct that ATM, MDC1, 53BP1, and SHLD2 are more sensitive to Zeocin than untagged parental U2OS cells. This indeed indicates that there is some functional impact of HaloTagging on one or more functions of these proteins. While ATM could not be tagged at either terminus, all the other proteins get robustly recruited to DNA damage sites as shown by their foci-forming ability. This largely indicates to us that the impact on function likely relates to events occurring post-recruitment (e.g. affected protein-protein interactions). While this is certainly an important limitation which we have now discussed in the revised manuscript, the types of experiments performed in this manuscript largely only depend on the ability of each protein to get recruited to break sites.

In terms of MDC1 the reviewers are correct that there is impaired function upon

HaloTagging, although these cells respond better to Zeocin treatment than their knockout counterparts. We do make a number of important conclusions from MDC1 and have included some limitations of HaloTag-induced changes in protein function in the Results section. Furthermore, we have included two new pieces of data in Figure S6 demonstrating that HaloTagged wild-type MDC1 can support HR as measured by the DR-GFP assay and that HaloTagged wild-type MDC1 also supports downstream recruitment of 53BP1 to DSB sites. Thus, while the HaloTagged MDC1 does have some sensitivity to Zeocin compared to wild-type cells, it can functionally support DSB repair.

It is important to note that transiently expressed H2B and NLS served as benchmarks for what to expect from a protein that should have a high fraction bound vs. a low fraction bound. However, because of the differences in how the proteins are expressed (transient overexpression vs. endogenous knock-in) it wouldn’t be fair to say much more than that MDC1 and H2B have similar fractions bound. While the fraction of bound MDC1 particles was higher than transiently expressed Halo-H2B (likely due to the transient overexpression of Halo-H2B), bound MDC1 (0.024 m^2^/s) particles had a higher diffusion rate than bound H2B (0.012 m^2^/s) which would not be unexpected. Conversely, free MDC1 diffused more slowly than free H2B which would be expected based upon the large difference in protein molecular weight. While we are not the first group to suggest that MDC1, through its PST repeat domain, constitutively interacts with chromatin (PMID: 31729360), we are the first to be able to directly observe constitutively bound MDC1 in living cells, rather than by chromatin-IP.

4) In the section: Functional validation of HaloTagged DDR proteins:In the absence of zeocin treatment, cells expressing Halo-MDC1 exhibit many spontaneous foci. Is it something known and if not, how can that be explained?

The reviewers are correct in their observation. While not all cells possess these foci without treatment, it is not unusual to observe low levels of foci in nearly all the tagged cell lines and this is not exclusive to Halo-MDC1. These spontaneous foci likely are indicative of low levels of endogenous DNA damage in cells that is bound by MDC1. DSBs occur in every cell of the human body every day, so it is not surprising to observe rapidly dividing cancer cells with small numbers of pre-existing foci in the absence of treatment. It is critically important, however, to point out that in all cases where there are some-pre-existing foci, that foci number is markedly increased upon Zeocin treatment, consistent with the retained ability of these tagged proteins to accumulate at sites of DSBs. A cursory inspection of the literature demonstrates that by immunofluorescence MDC1 can form foci in the absence of exogenous DNA damage (PMID: 18757370; 17158742).

Cells expressing ATM-Halo do not form foci after zeocin treatment: please comment.Concerning DNA-PK: this protein is highly abundant in the cell; however, it does not form foci after zeocin treatment. Is there an explanation? does it mean that even if the protein is very abundant, very few DNA-PK molecules are present within foci and sufficient for the next steps of NHEJ? Does it form a visible line after micro-irradiation?

We tagged ATM at both termini with HaloTag. Unfortunately, it was not functional when tagged at either terminus. Tagging one terminus interfered with proper localization in the nucleus (Figure 2A). Tagging the other terminus did not interfere with proper localization, however these cells were exquisitely sensitive to Zeocin similar to inhibiting ATM with an ATM kinase inhibitor (Figure S2C & D). This data indicated to us that HaloTagging ATM dramatically impairs function, so for these reasons it is not at all surprising that we do not observe foci in response to Zeocin-induced DSBs.

Concerning DNA-PKcs, we did not observe foci upon Zeocin treatment. This was expected based upon the well-known stoichiometry of DNA-PKcs during NHEJ where only two DNAPKcs proteins are bound at a DSB. Therefore, two molecules would not provide enough fluorescent signal above background to be able to observe this in densely labeled cells. In terms of response to laser-induced breaks, yes, DNA-PKcs does form a visible stripe (Quantified in Figure 4B & Supplemental Figure 4A; Movie provided in Supplemental Movie S1).

5) Page 12, line 206: It is stated that MDC1 has a higher protein abundance than ATM, SHLD1, SHLD2, and SHLD3 but one of the two MDC1 clones analyzed has the lowest protein abundance of all clones analyzed in this study (2400 molecules per cell according to the text and table I). Why are the two MDC1 clones differ so drastically from each other? Confusingly, the big difference between the two clones is not seen in the bar chart in Figure 3B and is not measured by flow cytometry.

Western blots for MDC1 are exceedingly challenging. We attempted to use multiple MDC1 antibodies and multiple transfer conditions. The differences between the two MDC1 clones were a result of somewhat inconsistent transfer onto nitrocellulose membrane. We have now included a statement to this effect in the revised manuscript. The difference in molecules/per cell visible on the bar chart in Figure 3B and the values obtained from flow cytometry experiments are also plotted in the bar chart.

6) Page 12, line 231: It is stated that the adjustment factors for DNA-PKcs are 0.62 and 0.79 but the application of these factors increases the molecule number for clone 1 but decreases it for clone 2 (the latter seems wrong).

We thank the reviewers for pointing out this discrepancy. Indeed, the values after TEV correction for DNA-PKcs C2 were input incorrectly. This has been corrected in the revised manuscript.

7) Single Particle Tracking analysis:Using analysis of single particle tracking, the authors can measure the diffusive properties of repair proteins. Diffusion is estimated in the presence and in absence of zeocin treatment. Thus, the cells contain many foci: all the traces in the nucleus are analyzed at once, inside and outside foci. The authors then used a 2 population model to fit the distribution of protein displacements.

The power of live-cell single-molecule imaging is that we are able to assess chromatin recruitment of individual DNA repair factors, even if the protein does not accumulate to form a DNA repair focus or prior to DNA repair focus establishment. For this reason, live-cell single-molecule imaging is conceptually very different from characterizing or tracking DNA repair foci and offers a completely new way of analyzing DNA repair by providing a comprehensive picture of how DDR proteins individually behave in both unperturbed conditions and after induction of DNA damage.

The SPT analysis allows the authors to give some interesting mechanistic insight but the authors could extract much more information from their data. Why are the values of Dslow not provided? The authors interpret the slow population as bound to damaged DNA. However, it is known that some proteins diffuse relatively fast inside repair foci, especially if they are able to form liquid-liquid phase separation. In addition, some proteins might diffuse slowly outside of foci, because of non-specific interactions (or even for MDC1 for example). Thus, it is possible that the slow molecules are a mixture of the molecules inside the foci with molecules exhibiting chromatin binding outside of the repair foci.

Thank you for this constructive suggestion. The *D*_bound_ data for each individual cell was originally included in the supplementary data, however, we have now moved this into Figure 5. Figure S5D contains the *D*_bound_ data compiled from each of three individual replicates.

Indeed, the reviewers are correct that one would potentially expect to observe differences in *D*_bound_ between factors that are physically incorporated in nucleosomes (e.g. H2B, *D*_bound_ 0.012±0.001 µm^2^/s), directly interact with nucleosomes or modified histone tails (e.g. MDC1, *D*_bound_ 0.024 ± 0.002 µm^2^/s), or form into liquid-liquid phase separated droplets (e.g. 53BP1, *D*_bound_ 0.099 ± 0.012 µm^2^/s). Our observations are consistent with those expectations, 53BP1 is 8-fold more dynamic in its bound state compared to H2B, which could be a consequence of rapid localized chromatin sampling, or movement of 53BP1 within a phase-separated droplet. These data are also consistent with new residence time analysis data we have included in the revised version of the manuscript where H2B has the residence time, which directly reports on the dissociation rate of the biochemical interaction resulting in the immobilization of the tracked molecule. In contrast, 53BP1 has the lowest residence time suggesting that it interacts with chromatin more dynamically.

There is no visualization of the traces allowing us to see if the slow molecules are indeed inside foci and the fast ones are outside. It would be essential to be able to see this for at least 1 cell for each repair protein.

As mentioned above, the strength of our approach is that we are able to analyze chromatin recruitment of the analyzed DDR factors without a requirement to form DNA repair foci. DNA repair foci likely represent DNA breaks that take particularly long to repair and require extensive processing. Therefore, rather than being limited to repair foci, our approach enables us to monitor individual DDR protein recruitment to DNA damage sites throughout the entire nucleus. To address the dynamics of a protein critical for DNA repair foci formation, we have included a new experiment to analyze 53BP1 at the single-molecule level and simultaneously detecting DNA repair foci. We demonstrate that long lasting static 53BP1 particles can be found inside and not associated with DNA repair foci, which suggests that chromatin association of 53BP1 is not limited to DNA repair foci, which further highlights the strength of our approach. In addition, this experiment demonstrates the 53BP1 molecules can rapidly transition in and out of DNA repair foci as well as transiently become immobile outside of DNA repair foci. These observations demonstrate that 53BP1 has two binding modes in DNA repair foci, likely representing molecules directly associated with chromatin and recruited through phase-separation based interactions, respectively.

Are mixed traces observed, with a slow and a fast part?Is it possible to estimate the residence time of each protein inside repair foci or on their substrate?

Thank you for the suggestion. As described above we do observe 53BP1 molecules that transition between free and bound states. Reliable quantification of the transition dynamics requires highly accurate particle tracking over extended periods of time and is not feasible with the data generated in this work. A particular issue in eukaryotic cells is the drift of molecules out of the focal plane of the experiment making it challenging to estimate true transition rates. We attempted to analyze the residence time of 53BP1 particles in DNA repair foci using the dual labeling strategy described above. Unfortunately, the number of molecules associated with DNA repair foci is limited making precise quantification beyond the anecdotally observed long binding events challenging. Future studies that are specifically geared towards addressing this question will be carried out.

Are the proteins inside foci exchanging with the rest of the nucleus or are they stuck inside the focus during the entire trace?Since you use bright JF, it should be possible to have long traces: the authors should show a distribution of the traces' length for each repair protein.

As described above we do observe 53BP1 molecules that transition in an out of DNA repair foci. We analyzed the residence time (equivalent to the track length) for static particles of all proteins which approximates the dissociation rate of the underlying biochemical interaction. We observed an increase in the residence time for 53BP1, SHLD2, and SHLD3 after zeocin treatment suggesting that the biochemical basis of the interaction of these proteins with chromatin is changed by DNA damage induction (for example by chromatin modification or break resection). The observed residence times are limited in length by photobleaching even though we are using bright and photostable JF dyes. Future studies with modified imaging conditions will be necessary to more accurate determine the total binding time of the tagged DNA repair factors.

Do you see a change in protein diffusion in the absence of zeocin treatment and in the presence of zeocin treatment outside of foci?

Because of the nature of live-cell single-molecule imaging, we were not limited to studying DNA repair foci, but were able to use a nucleus-wide approach to monitor how individual DDR proteins. For these experiments, we did not specifically use a marker for DNA repair foci, so this analysis cannot be performed with these particular data sets because there is no reference for where repair foci are located in the nucleus. This type of analysis could indeed be performed in the future but we think this type of analysis is outside the scope of the current manuscript. However, we have included one experiment in Figure S5 where we labeled Halo-53BP1 with JFX650 at the single-molecule level and densely labeled with JF503 to mark 53BP1 foci where demonstrate dynamic associations of 53BP1 inside and outside foci.

The authors also use H2B and NLS as controls. The values obtained should be compared with values found in the literature.

This is a great suggestion. We have included a section in the revised text where we reference values for Halo-H2B and Halo-NLS in the manuscript published in *eLife* which describes SpotOn analysis by Hansen *et al.* (PMID: 29300163). For both proteins the fraction of bound particles is comparable: Halo-H2B F*_Bound_* = 66% vs. ~75% in Hansen *et al.;* 3XFLAG-Halo-3XNLS F*_Bound_* = 16% vs. ~10-15% in Hansen *et al.,* but it is important to note that the Halo-3XNLS described in Hansen *et al.* differs from that used in this manuscript due to the addition of a 3X FLAG tag.

For Halo-H2B we report a D*_Free_* of 2.164 ± 0.194 µm^2^/s vs. ~4 µm^2^/s in Hansen *et al.* Similarly, for 3xFLAG-Halo-3xNLS we report a D*_Free_* of 3.784 ± 0.512 µm^2^/s vs. ~10 µm^2^/sec in Hansen *et al.* The reported values for the D*_Free_* differ between those reported in this manuscript and those reported in Hansen *et al.* for a couple reasons. Hansen *et al.* performed stroboscopic photo-activation single particle tracking (spaSPT) in which photoactivatable JF dyes allowed for imaging of ~1 particle per frame on average, while we did not use photoactivatable dyes and had a higher labeling density (~20 ± 5 localizations per frame for Halo-H2B and ~11 ± 1 localizations per frame for Halo-NLS). To determine the free diffusion coefficient of extremely rapidly diffusing molecules like Halo-NLS low labeling density is critical because the tracking algorithm is more likely to connect molecules in close proximity and the likelihood of inadvertently connecting particles incorrectly increases with the rate of diffusion and the labeling density. Importantly, while the increased labeling density in our experiments can impact the absolute value of the free diffusion coefficient of rapidly diffusing molecules, relative diffusion coefficient values are highly reliable. In addition, for the shieldin complex where conclusions were drawn based upon the free diffusion coefficient of the individual subunits labeling density in the nucleus was comparable and very low, which makes these measurements highly reliable.

Finally, it is not clear how the histones H2B are tagged in this study: is it also an endogenous H2B-Halo tag? Is it a stable cell line but not endogenous, or is it a transient transfection?

Both Halo-NLS and Halo-H2B were expressed by transient transfection.

8) Laser micro-irradiation induces massive damage and may not be reflective of physiological encountered DNA damage. Have the authored considered using Cas9-induced DSBs as a defined and targeted DNA damage? I understand that adding such experiments for all proteins would be a massive endeavor, but maybe this could be done for MDC1 and/or the Shieldin complex. Regardless, the limitations of the laser micro-irradiation approach should be discussed.

The application of a versatile tag like HaloTag to study DNA repair using experiments with either sparse or dense protein labeling opens up many exciting opportunities for future research. While we have discussed performing very fast CRISPR on demand, we agree with the reviewers that this would be a massive endeavor and would argue that the addition of these types of experiments are outside the scope of the current manuscript. We have now included a brief discussion in the Results section describing some limitations of using laser micro-irradiation for recruitment analyses.

9) Figure 6 lacks data for the analysis in the presence of zeocin, in the way it was done in the analysis for Figure 5. Such data will corroborate the foci analysis and potentially reveal differences in the recruitment of MDC1 to damaged sites.

We agree with the reviewers and this is an excellent suggestion. We have included single particle tracking of the three different MDC1 mutants in the presence/absence of Zeocin (Figure S6C). The data demonstrates that while the bound fraction of full-length and the BRCT deletion mutant do not change upon Zeocin treatment, there is a significant increase in the bound fraction for the PST deletion mutant, complementing our other observations that MDC1 can interact in at least two ways with chromatin. Furthermore, this new data demonstrates that the PST domain is critical for supporting MDC1’s constitutive association with chromatin, but is not required for MDC1 recruitment to DNA DSBs by its H2AXbinding BRCT domains.

10) What is the evidence that a single molecule can be tracked in Figures 5 and 6, as opposed to a single particle that may be composed of multiple proteins?

For these experiments we label only a small fraction of the overall cellular pool of each protein. It is entirely possible that proteins that are part of larger multi-protein or homomultimers are labeled. These could potentially be distinguished of the individual proteins and larger complexes have distinct diffusion properties. Importantly, the confidence that the vast majority of signals are derived from a single fluorophore is very high. To demonstrate this point, we have included intensity profiles of the analyzed trajectories for MDC1 and RIF1, which are highly static reducing the probability of intersecting tracks leading to additive intensity values.

11) Page 13, line 256: It was surprisingly observed that REV7 and SHLD3 have vastly different recruitment times to laser-induced damage although both factors are known to interact. At the end of the Discussion section, it is speculated that this might reflect REV7's role in TLS but the reader would benefit if such an interpretation was offered earlier in the paper. Moreover, if this interpretation was correct, one would expect that REV7 was recruited to laser damage independently of SHLD3. This should be tested by siRNA-mediated depletion of SHLD3 (or other factors upstream of SHLD3).

These data do not directly report on the absolute order of recruitment of factors to DSBs but rather only report on the relative time to maximal accumulation of each factor to laser-induced DSBs. This particular question becomes much more complicated by issues related to stoichiometry of each factor at breaks. Additionally, because each cell line is normalized to its own peak fluorescence intensity and because laser power was not equally applied for all factors due to dramatic differences in overall protein expression, it would not be appropriate to make a direct comparison between proteins in terms of the absolute number of proteins recruited to laser-induced DSBs. To avoid overinterpreting our results we have removed the more speculative statements regarding the interplay of REV7 and SHLD3 in recruitment to DSBs.

12) Page 14, lines 260-263: It is observed that SHLD1 is not recruited to laser damage although it forms foci after zeocin treatment. The authors then speculate that this might suggest that SHLD1 recruitment is the key regulatory step for a fill-in reaction. This is not clear. Are the authors suggesting that fill-in takes place at zeocin-induced breaks but not at laser-induced breaks? Please clarify.

When we initially performed the laser micro-irradiation experiments for SHLD1, we took images every thirty seconds after irradiation and never could detect SHLD1 recruitment up to ~90 minutes post-irradiation. At the time, we made this speculation to better make sense of this unexpected data. However, we repeated this experiment again using a much longer period of time between images (10 minutes) and successfully detected recruitment of SHLD1 to laser-induced DNA breaks (Figure S4D). This new data suggests our initial imaging conditions were leading to fluorophore bleaching prior to SHLD1 accumulation which impaired our ability to detect it due to low protein abundance in combination with a low amount of protein recruited to break sites. SHLD1 is recruited, but in very low amounts to DSBs which we could visualize, but not easily quantify. This new data is now included in Figure S4D and we have modified the text to account for the new data.

13) Page 14, line 266: The authors state that the measured recruitment kinetics provide insight into the interdependencies of the shieldin complex components. As evidenced by the very early recruitment kinetics of REV7 (possibly due to its role in TLS), this may not be true since the factors could have roles in other processes. Thus, the evaluation of interdependencies requires the measurement of the recruitment kinetics in situations when individual components of the shieldin complex are depleted by siRNA technology. Thus, the authors should assess the recruitment kinetics in cells depleted for SHLD2/3, REV7, or 53BP1/RIF1.

We have changed the wording here in the Results section and in the discussion. This data in its current state can only report on the overall kinetics of Shieldin assembly. We are by no means challenging the well-documented genetic dependencies in Shieldin complex assembly which has been described by several other groups (PMID: 30022168, 29656893, 30022119).

14) Page 17, line 319: SHLD1 is not recruited to chromatin after DSB induction despite its low abundance, and the authors state that this is consistent with its lack of recruitment to laser damage. However, it is clearly shown to form foci at DSB. Moreover, how does this finding fit the author's suggestion from above that SHLD1 is the key regulator of the fill-in reaction?

When we initially performed the laser micro-irradiation experiments for SHLD1, we took images every thirty seconds after irradiation and never could detect recruitment up to ~90 minutes post-irradiation. At the time, we made this speculation to better make sense of this unexpected data. However, we repeated this experiment again using a much longer period of time between images (10 minutes) and successfully detected recruitment of SHLD1 to laser-induced DNA breaks (Figure S4D). This new data suggests our initial imaging conditions were leading to fluorophore bleaching prior to SHLD1 accumulation. SHLD1 is recruited, but in very low amounts to DSBs which we could visualize, but not easily quantify. This new data is now included in the manuscript in Figure S4D and we have modified the text to account for this new data which is consistent with that observed in live-cell imaging of SHLD1 foci.

15) The authors try to address the unexpected findings for SHLD1 (no recruitment to laser damage and no increase in the chromatin-bound fraction after zeocin treatment) in the discussion on page 22, last paragraph, and suggest that SHLD1 may either not bind every DSB where SHLD2 is present or may reside at a DSB in lower copy numbers than SHLD2. Both explanations appear inconsistent with the robust formation of SHLD1 foci at zeocin-induced DSB, the first suggestion can easily be tested with co-localization experiments using HaloTag SHLD1 and SHLD2 ab or vice versa.

We agree with the reviewers that this or a comparable experiment would be critical considering the discrepancy between SHLD1 foci formation after Zeocin and the apparent lack of SHLD1 recruitment to laser micro-irradiated sites. However, the new data presented in Figure S4D where we demonstrate successful detection of SHLD1 recruitment to laser micro-irradiated sites helps to resolve the speculation originally in the text to explain the difference between the two experiments. We have now corrected the text in accordance with the new data.

16) Page 21, line 390: The authors suggest the model that RNF169 delays 53BP1 recruitment to DSBs. Although this interpretation is consistent with the presented data, the model should be tested by 53BP1 recruitment measurements after siRNA-mediated depletion of RNF169.

In the revised version of the manuscript we removed sections where we made bold conclusions regarding the differences between RNF169 and 53BP1 recruitment times to laser micro-irradiated sites. Additional studies are being planned to go further in depth into the interplay between RNF169 and 53BP1 in DSB repair in the future.

[Editors’ note: what follows is the authors’ response to the second round of review.]

The manuscript has been improved but there are some remaining issues that need to be addressed, as outlined below:[…]The manuscript is well-written, but some literature reference and information in the methods section are missing.In conclusion, this is an interesting study that applies novel microscopy techniques to DNA double-strand break repair proteins. However, the study remains somewhat descriptive, which limits the mechanistic insight gained from the study and the analysis remains incomplete in several instances for lack of genetic corroboration of the main conclusion about the Shieldin complex recruitment.Recommendations for the authorsEssential revisions1) Line 144: The authors conclude that "the HaloTag does not impact the proper cellular localization of these proteins" based on fluorescence microscopy of the Halo-tagged proteins after JF646 labeling. This conclusion cannot be made, because it would require examination of the localization of the untagged proteins under the same conditions. Please qualify your statement.

In the revised manuscript we have now included an additional statement to qualify this point.

2) Line 146: The authors conclude that "HaloTagging ATM at the N-terminus led to nuclear exclusion". The authors cannot make this conclusion without imaging the untagged ATM under the same condition. Please qualify your statement.

We thank the reviewers for this comment. We HaloTagged ATM at both the C-terminus and N-terminus. While the C-terminally tagged ATM was expressed predominately in the nucleus, the N-terminally tagged ATM was largely excluded from the nucleus. This suggests that tagging one of the termini affects normal ATM subcellular localization. ATM has been long known to be largely localized to the nucleus (Gately *et al.* Mol Biol Cell, 1998 PMID: 9725899), so it would seem reasonable to conclude that the N-terminal HaloTagging of ATM is what is responsible for its nuclear exclusion even without imaging the untagged protein in living cells. However, we have qualified this statement so as not to state unequivocally that HaloTagging ATM at the N-terminus leads to nuclear exclusion of ATM.

3) Line 183: It cannot be concluded that "most possess full DNA repair functionality" unless the cell lines expressing Halo-tagged proteins are compared to their gene knockout counterparts. This was only performed for 53BP1 and MDC1, where partial functionality was observed. Please qualify your statement.

In the revised version of the manuscript, we have reworded and qualified this statement by stating that these data suggest that most of these proteins retain at least partial DNA repair functionality.

4) Page 13, line 256: It was surprisingly observed that REV7 and SHLD3 have vastly different recruitment times to laser-induced damage although both factors are known to interact. At the end of the Discussion section, it is speculated that this might reflect REV7's role in TLS but the reader would benefit if such an interpretation was offered earlier in the paper. Moreover, if this interpretation was correct, one would expect that REV7 was recruited to laser damage independently of SHLD3. This should have been tested by siRNA-mediated depletion of SHLD3 (or other factors upstream of SHLD3) and this limitation should be explicitly acknowledged in the text.

In the revised manuscript we have added a statement explicitly acknowledging this limitation in the text.

5) Page 14, line 266: The authors state that the measured recruitment kinetics provide insight into the interdependencies of the shieldin complex components. As evidenced by the very early recruitment kinetics of REV7 (possibly due to its role in TLS), this may not be true since the factors could have roles in other processes. Thus, the evaluation of interdependencies requires the measurement of the recruitment kinetics in situation when individual components of the shieldin complex are depleted by siRNA technology. Thus, the authors should assess the recruitment kinetics in cells depleted for SHLD2/3, REV7 or 53BP1/RIF1. The caveat of roles in independent processes should be explicitly mentioned.

In the revised manuscript we have added a statement explicitly acknowledging this limitation in the text as well as describing the caveats of roles of these factors in independent processes.

6) Line 293: It is counter-intuitive that SHLD2 foci depends on SHLD3, which is recruited to foci significantly later than SHLD2. The two-step model suggested in the Discussion to explain this observation is not supported well by the data, because one would expect the initial (pre-SHLD2) step of SHLD3 to also be detected in the LMI experiments. Further, SHLD2 actually dissociates when SHLD3 associates with laser stripes, which is not discussed by the authors. It would strengthen the manuscript if the model could be tested experimentally by the authors.

It is important to note that for these experiments each protein was normalized to its maximal accumulation. To avoid having to normalize the fluorescence intensities, we performed laser micro-irradiation and imaged SHLD2 and SHLD3 using the exact same imaging conditions so we could directly compare between these two proteins and measured absolute fluorescence intensities as a readout of recruitment (Figure S4D). These experiments demonstrate that SHLD3 and SHLD2 are simultaneously recruited to DNA breaks. However, SHLD2 reaches maximal accumulation much earlier than SHLD3 which continues to accumulate in excess of SHLD2 (Figure S4D). In terms of the appearance that SHLD2 dissociates when SHLD3 associates, this appears to be driven by 3-4 cells where there was a reduction in SHLD2 at laser-induced DNA damage sites, while most cells did not have this appearance (Figure S4A and S4D).

7) A similar study was conducted previously (Aleksandrov et al. 2018), which reported recruitment half-times to laser stripes significantly different than the current study for some proteins and in other cases similar half-times. For example, the recruitment times for MDC1, RNF168, RNF169, and 53BP1 was reported by Aleksandrov to be 35s, 78s, 203s, and 307s, where the current study report 77s, 69s, 186s, 669s. It would be in place to reference the previous study and compare findings.

This is an excellent suggestion from the reviewers. We have now included a comparison of our findings to those in Aleksandrov *et al.* While our results for RNF168 and RNF169 essentially match those in Aleksandrov *et al.* Our findings for MDC1 and 53BP1 differ ~2-fold. While our experiments used endogenously HaloTagged human 53BP1, Aleksandrov *et al.* used a HeLa cell line expressing mouse 53BP1, this could provide a reasonable explanation for the differences between the two studies. Additionally, the cell lines we engineered exclusively express HaloTagged proteins while the cell lines used in Aleksandrov *et al.* express eGFP-tagged proteins in the presence of the wild-type unedited proteins. Therefore, the differences in the recruitment of MDC1 and 53BP1 between the two studies could be a consequence of differences in the protein sequence, gene dosage or competition with the endogenous protein leading to altered recruitment in Aleksandrov *et al.* compared to what we observe in the present study. We discuss these differences in the revised manuscript.

8) It is surprising that Halo-H2B only displays 66% chromatin binding. The assumption is that JF646 binds irreversibly to the Halo-tag, but is it possible that some free JF646 is present and gives rise to the "free" pool of fluorophores? This could easily be tested by formaldehyde fixation which should give 100% chromatin binding if no free JF646 is present.

It is important to note that Halo-H2B was transiently overexpressed in the presence of wildtype H2B, so it is not unreasonable to assume that the cellular pool of H2B is higher leading to a competition of H2B and Halo-H2B competing for inclusion into a limited number of nucleosomes and therefore the values reported here may artificially lower than expected. Despite this, the value we obtained for Halo-H2B in terms of the fraction of bound particles (~66%) is comparable to that described in Hansen *et al.* (Hansen *et al., eLife.* 2018) where they observed an F_bound_ of ~75% using Spot-On analysis and transiently expressing Halo-H2B using the same plasmid*.* The purpose of using the Halo-H2B was only to have a control for what to expect to observe if a protein is largely chromatin bound as opposed to the 3x-FLAGHalo-NLS which was used as a control for the lower bound of what one would expect if a protein was largely freely diffusing. We do not believe free fluorophores contribute to our analysis, since we do not observe nuclear particles in control cells that do not express a Halotagged protein.

9) Line 331: A two-state diffusion model is assumed where particles either freely diffuse or are chromatin bound. How would the conclusions be affected if a third state was allowed where a protein is part of a slow diffusing macromolecular complex.

Inclusion of a third state typically has a minimal effect on the static fraction of proteins we have studied in the past. The third state typically has intermediate mobility and comes at the expense of the fraction of freely diffusing molecules. In our past work on telomerase, we have explicitly used a three-state model because the TERT protein can be either freely diffusing or part of the large telomerase RNP, which significantly reduces its mobility (Klump et al. Cell Reports 2023). Similarly, we have used a three state model to analyze 53BP1 trajectories inside and outside of DNA repair foci in this updated manuscript, assuming that 53BP1 can be either freely diffusing, directly bound to chromatin, or part of phase-separated DNA repair foci.

10) Figure 5: The scatter plots should be displayed as "super plots" where data points for each of the 3-4 independent experiments are presented in different colors/symbols (Lord et al. 2020). This would reveal any systematic differences between experiments. For example, it looks like RNF169 data points can be divided into two populations.

We have now plotted these plots as super plots as the reviewers have suggested in the revised version of the manuscript.

11) Page 21, line 390: The authors suggest the model that RNF169 delays 53BP1 recruitment to DSBs. Although this interpretation is consistent with the presented data, the model could be tested by 53BP1 recruitment measurements after siRNA-mediated depletion of RNF169. This limitation should be explicitly acknowledged in the text.

While we had suggested in the original submission that RNF169 delays 53BP1 recruitment, this language was removed in the second submission. It is not clear to us, which statement the reviewer is referring to.

12) The sources of several plasmids are missing e.g. pX330 in line 665.

The sources for px330, pFastBac Dual, and pRK2 have now been included in the Material and Methods section.

13) A table with oligonucleotides and gRNAs used in the study should be included.

A table with oligonucleotides and gRNAs used in this study has been included in the resubmission of this manuscript.

14) Line 511: Which data allow the authors to conclude that the PST domain of MDC1 "facilitates DDR signal amplification"?

We admit that this may have been strongly worded. Therefore, we have removed the term “signal amplification” and kept the statement that it is important for DSB repair based upon results from the HR DR-GFP assay.

15) The authors conclude that MDC1 and RIF1 are constitutively associated with chromatin. If this is the case then one might expect the localization of MDC1 and RIF1 to follow that of condensed chromosome when cells progress from interphase into mitosis. Indeed, there is evidence for this for RIF1 (Watts et al. 2020), but for MDC1 I could not find such evidence in the literature. However, I suspect that the authors in their data have images of mitotic cells that could answer this question.

We thank the reviewers for this thoughtful comment. This is actually an exciting area of ongoing inquiry that is nearing completion and submission for publication. Because of this we feel that presenting data related to this question is outside of the scope of the current manuscript.

16) To quantify the mobility inside foci versus outside foci, and the flux of proteins in and out of foci, would it be possible to make a density map of all the repair proteins in the nucleus? Using this density map, the foci appear clearly, and it is then possible to separate trajectories within foci from those outside foci.

We have added a new analysis that filters trajectories using a mask of the nuclear 53BP1 foci. Trajectories are separated into two groups, one in which trajectories overlap with DNA repair foci and a second group of tracks that never co-localizes with repair foci and analyzed the data using a three stated model assuming 53BP1 can be freely diffusing, statically bound to chromatin, or part of phase separated DNA repair foci (which would move with an intermediate diffusion coefficient). In DNA repair foci 53BP1 particles moving with an intermediate diffusion coefficient were enriched compared to particles that did not colocalize with DNA repair foci, consistent with the hypothesis that 53BP1 can be recruited to DNA repair foci by a phase separation mechanism.

17) Figure S5B: After zeocin treatment: is it possible that the lines between foci are misslinking?

In the context of single particle tracking miss-linking can occur, for instance if a static particle is failed to be detected in a given frame due to signal fluctuations (for example as a result of photo-blinking) a trajectory can be incorrectly linked to a nearby molecule. In general, the number of incorrect steps included in our analysis is minimal compared to correct linkages and therefor do not impact global step distance analysis like we do with the Spot-On tool. Correct linkage is extremely important for residence time analysis because long binding events can be chopped up into short trajectories. For this reasons, we averaged multiple frames together to amplify static binding events and minimize the contribution of signal fluctuations.